# Effectiveness of Oregano and Thyme Essential Oils as Alternatives for Sulfur Dioxide in Controlling Decay and Gray Mold and Maintaining Quality of 'Flame Seedless' Table Grape (*Vitis vinifera* L.) during Cold Storage

Usama K. El-Abbasy [1], Mohamed A. Abdel-Hameed [1], Harlene M. Hatterman-Valenti [2,*], Ali R. El-Shereif [3] and Ahmed F. Abd El-Khalek [1,*]

1 Department of Horticulture, Faculty of Agriculture, Tanta University, Tanta 31527, Egypt; usama.elabbasy@agr.tanta.edu.eg (U.K.E.-A.); mohamedadel0109249@gmail.com (M.A.A.-H.)
2 Department of Plant Sciences, North Dakota State University, Fargo, ND 58108, USA
3 Department of Horticulture, Faculty of Agriculture, Kafrelsheikh University, Kafr El-Sheikh 33516, Egypt; aelshereif@agr.kfs.edu.eg
* Correspondence: h.hatterman.valenti@ndsu.edu (H.M.H.-V.); ahmed.gameal@agr.tanta.edu.eg (A.F.A.E.-K.); Tel.: +2-040-345-5584 (A.F.A.E.-K.)

**Abstract:** The current study was carried out over two seasons (2020 and 2021) to assess the effects of preharvest treatments with oregano and thyme essential oils (EOs) as an alternative to the traditional use of sulfur dioxide ($SO_2$) during cold storage of grape clusters cv. Flame Seedless. Grapevines were sprayed with oregano or thyme essential oils at 2000 or 4000 µL/L two days before harvest. The results confirmed that oregano and thyme EOs treatments reduced the physiological loss in weight, decay incidence, gray mold, rachis browning index, and berry shattering as compared to sulfur dioxide and untreated fruits. In addition, EOs had higher marketable percentage, firmness, and visual appearance cluster scores, while they reduced the deterioration in titratable acidity (TA) and ascorbic acid (AsA) contents, slowing the increases in soluble solids content (SSC) and SSC/TA ratio of berries, and improving total anthocyanin content. Moreover, these EOs delayed berry activities of polyphenol oxidase (PPO), peroxidase (POX), and pectin methylesterase (PME) enzymes during cold storage. Results suggest that preharvest application with either oregano or thyme EOs at 2000 µL/L might be a promising eco-friendly and safe candidate as an alternative to conventional $SO_2$ used to control decay incidence and gray mold rot caused by *Botrytis cinerea*, and the EOs were effective in maintaining the quality of grape clusters during cold storage for up to 45 days.

**Keywords:** *Vitis vinifera*; flame seedless; oregano oil; thyme oil; preharvest; cold storage; decay; gray mold; *Botrytis cinerea*; berry quality; polyphenol oxidase; peroxidase; pectin methylesterase

## 1. Introduction

Grape (*Vitis vinifera* L.) is one of the most economic and important fruit crops around the world, ranking fourth behind bananas, watermelons, and apples [1], and Egypt's second most important fruit after citrus [2]. In 2020, the global total harvested area of grapes was 6.95 million hectares, with a total production of 78 million tons, while Egypt's total harvested area was 71.9 thousand hectares, with a total production of 1.6 million tons. Approximately half of the global grape production is consumed as wine, with the other half consumed as raisins, juices, direct consumption, and jams [3]. Grapes are a good source of antioxidants and numerous active biological compounds (i.e., phenolic acids, flavonoids, anthocyanins, stilbenes, and lipids) that have been shown to play a protective role against cardiovascular disease, arteries, infections, cancers, eye disorders, diabetes, obesity, and nervous system functions [4].

Grapes are vulnerable to a variety of postharvest diseases caused by various pathogens which significantly reduce their commercial value and edible quality [5]. More than 12% of postharvest losses for global crops are typically attributable to fungi [6]. Scientists estimated that fungal decay losses comprise 10–40% of the world's total grape production [7], making fruits' postharvest diseases the most serious cause of fruit production loss. Postharvest diseases in table grapes are caused by various fungi such as *Aspergillus niger*, *Alternaria alternata*, *Colletotrichum gloeosporioides*, *Lasiodiplodia theobromae*, *Penicillium digitatum*, *Phomopsis viticola*, *Rhizopus stolonifer*, and *Botrytis cinerea* [8]. However, gray mold, caused by *Botrytis cinerea*, is one of the most damaging diseases, affecting approximately 90% of table grape production during postharvest, handling, storage, and marketing [9,10].

Various synthetic fungicides have been used to inhibit pathogenic fungi attacks on plants; however, the use of such fungicides results in stimulating the evolution of fungicide-resistant strains, environmental pollution, and negative human health effects [11,12]. Table grapes are commercially fumigated with sulfur dioxide ($SO_2$) to control postharvest decay derived from *Botrytis cinerea* fungi during cold storage. Unfortunately, the necessary concentration may trigger phytotoxic symptoms such as rachis browning, sulfur taste, hairline cracks on berry skin, bleaching, discoloration, and sulfide residues that may cause hypersensitivity reactions in some people [13–16]. The amount of $SO_2$ residue in grapes after fumigation differed among various parts of the berry and could be ranked from low to high as follows: pulp, skin, brush, rachis, and pedicel [17]. Therefore, the use of $SO_2$ gas on table grapes is not allowed on organic grapes [14].

To control gray mold in table grapes, several eco-friendly alternatives to $SO_2$ use have been reported, including the use of ultraviolet irradiation (UV-C), ozone ($O_3$), hot water or vapor heat treatments, carbon dioxide ($CO_2$) or oxygen ($O_2$) shocks, nanocomposite treatments, biological control agents, natural salts, and essential oils [18–25].

Essential oils (EOs) present in plants are natural antioxidants with antimicrobial and biodegradable properties and have no residual effect on fresh produce [26–30]. Terpenes (monoterpenes and sesquiterpenes), terpenoids (isoprenoids), and aliphatic and aromatic compounds such as aldehydes and phenols are primarily responsible for EOs' antifungal properties. Terpenes are naturally occurring hydrocarbons with a variety of chemical and biological properties that account for up to 90% of the most important oil components [31]. Additionally, EOs increase the total phenol content in fruits which plays a key role in plant resistance and acts as a defense mechanism against the invasion of plant pathogens [32], and they have beneficial effects on fruits' physio-biochemical attributes [33,34].

Compounds such as thymol, carvacrol, and eugenol have a proven fungicidal activity, and essential oils rich in these components were shown to have the highest inhibitory activity against various postharvest pathogens [35–37]. Thymol and carvacrol are monoterpenes present in many EOs, including those from oregano [*Origanum vulgare* L.] and thyme [*Thymus vulgaris* L.]; they may inhibit fungal growth by stimulating plant defense responses and inhibition of the target of the rapamycin (TOR) pathway [38]. Oregano oil was found to have strong antifungal properties, completely inhibiting fungal growth in vitro [39] and controlling gray mold in vitro and in vivo for table grapes without adverse effect on quality [40]. Thyme oil was reported to control a variety of fungal pathogens including *Botrytis cinerea* on tomato and on avocado during in vitro tests [41,42], in vivo on Satsuma mandarins [*Citrus unshiu* Marc.] and apple [*Malus domestica* (Suckow) Borkh.] [43,44], and during storage on papaya [*Carica papaya* L.] [45].

Previous studies investigated the antimicrobial efficacy of EOs (including thyme and oregano) as a postharvest application, and when tested in vivo, the fruits were artificially wounded and inoculated. Furthermore, to the best of our knowledge, the efficacy of such EOs has not been studied in comparison to $SO_2$. Therefore, this study aimed to evaluate the effects of preharvest application of thyme and oregano essential oils as eco-friendly $SO_2$ alternatives to control decay, reduce gray mold disease caused by *Botrytis cinerea*, and maintain the fruit quality of 'Flame Seedless' grapes during cold storage.

## 2. Materials and Methods

### 2.1. Source and Analysis of Essential Oils

Essential oils of oregano and thyme purity (100%) were purchased (EL-Masrayia Company for Natural Oils, Cairo, Egypt) and stored in opaque bottles at 4 °C until further use. These oils were analyzed at the Central Laboratory, National Research Center, Giza, Egypt before being used (Figure 1). The essential oil components were identified by GC-MS (Hewlett-Packard 5890 A series 11, Agilent Technologies, Palo Alto, CA, USA) equipped with a flame ionization detector (FID) fused silica capillary column (50 m long × 0.25 mm internal diameter, film thickness 0.32 μm) coated with DB-5 (5% phenyl, 95% methyl polysiloxan) to separate the different volatile components. The oven temperature was programmed from 50 °C to 260 °C at a rate of 3 °C per min and held at 260 °C for five minutes. Helium was used as a carrier gas at a flow rate of 1 mL/min. The quantification of all the identified components was investigated using a percent relative peak area. The components of each oil were separated, and the chromatogram obtained was identified by comparing the mass spectra to those from the National Institute of Standards and Technology (NIST) library. Also, all used chemicals in the current research were imported from Sigma Aldrich, St. Louis, MO, USA.

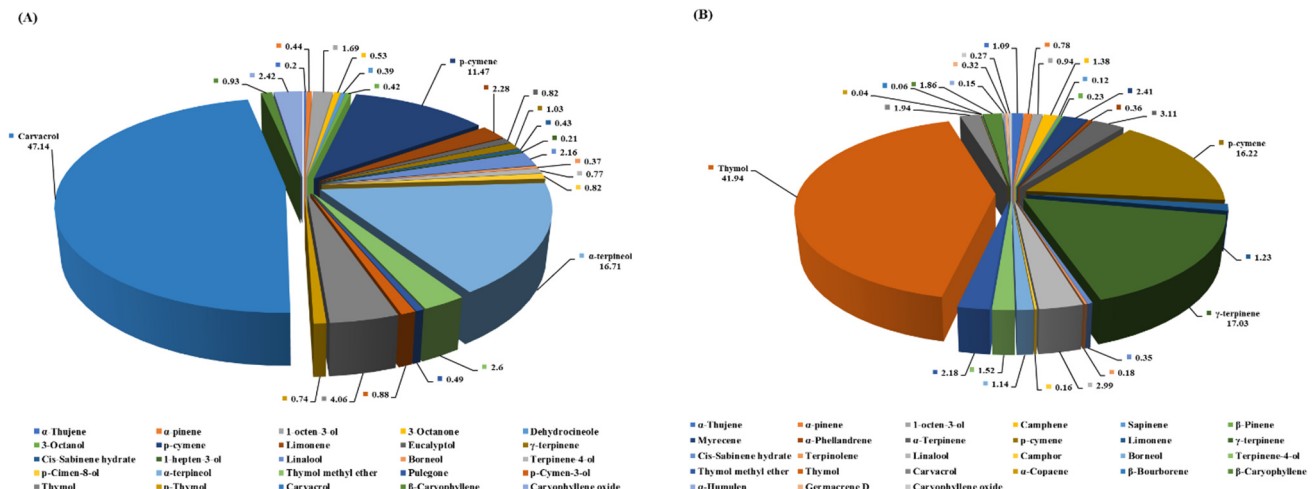

**Figure 1.** Chemical composition of oregano (*Origanum vulgare* L.) (**A**) and thyme (*Thymus vulgaris* L.) (**B**) essential oils by GC-MS analysis.

### 2.2. Plant Materials and Experimental Procedure

The present study was conducted during the 2020 and 2021 seasons, on 10-year-old 'Flame Seedless' grapevines grown in El-Masria orchard, Bazel Marco company, Markaz Badr distract, Modriat El-Tahrir, El-Beheira Governorate, Egypt (latitude, 30°57′ N; longitude, 30°71′ E). The grapevines were planted in a row spacing of 2.5 m with 3 m between rows in sandy soil under a drip irrigation system and subjected to all ideal agriculture practices as recommended by the Egyptian ministry of agriculture and land reclamation.

Sixty grapevines approximately uniform in size and free of visible symptoms of infection were sprayed with oregano or thyme essential oils (0, 2000 and 4000 μL/L) and tween-80 as a surfactant 48 h before the clusters were picked during the second week of July in both seasons. Each vine cluster was sprayed until the solution dripped from the fruit. The picked clusters were placed in plastic boxes (5 kg capacity) and transferred by refrigerated vehicle to El-Eman packing house for sorting, packing, cooling, and freezing agricultural crops (Markaz Badr, El-Beheira Governorate, Egypt). Upon arrival, clusters were sorted; damaged and injured berries, as well as non-homogeneous berries, were excluded. After that, 72 clean randomly selected clusters from each treatment were divided into four storage periods (0, 15, 30, and 45 days). Each storage period consisted of three replicates and each replicate contained six clusters. All clusters were placed in polyethylene terephthalate

(PET) lid punnet boxes (one cluster/punnet box) with dimensions of $19 \times 12 \times 8.5$ cm. The punnet boxes were placed as a single layer inside corrugated boxes with dimensions of $60 \times 40 \times 9$ cm. Only the control clusters were divided into two groups; the first one was preharvest-treated with distilled water and the second one was treated by $SO_2$ generator dual-release pads that contain sodium metabisulfite (PROTEKU®, Calle C, esquina calle F-03109 TIBI, Alicante, Spain) during the cold storage period. Thus, the experiment included six treatments: (1) control (preharvest-sprayed with distilled water), (2) $SO_2$, (3) 2000 µL/L oregano oil, (4) 4000 µL/L oregano oil, (5) 2000 µL/L thyme oil, and (6) 4000 µL/L thyme oil. All experimental boxes were covered with polyethylene sheets 0.04 mm thick and subjected to pre-cooling by placing them into a forced air-cooling room (pressures 0.6 to 7.5 mbar and air flows 0.001 to 0.003 m$^3$/second/kg) at 2 °C for four hours. After that, the boxes were stored at 1 °C and 90–95% RH for 45 days. The fruits' physical and chemical characteristics were measured at harvest and then every 15 days from the start of the cold storage period to the end. The experimental procedures are shown in Figure 2.

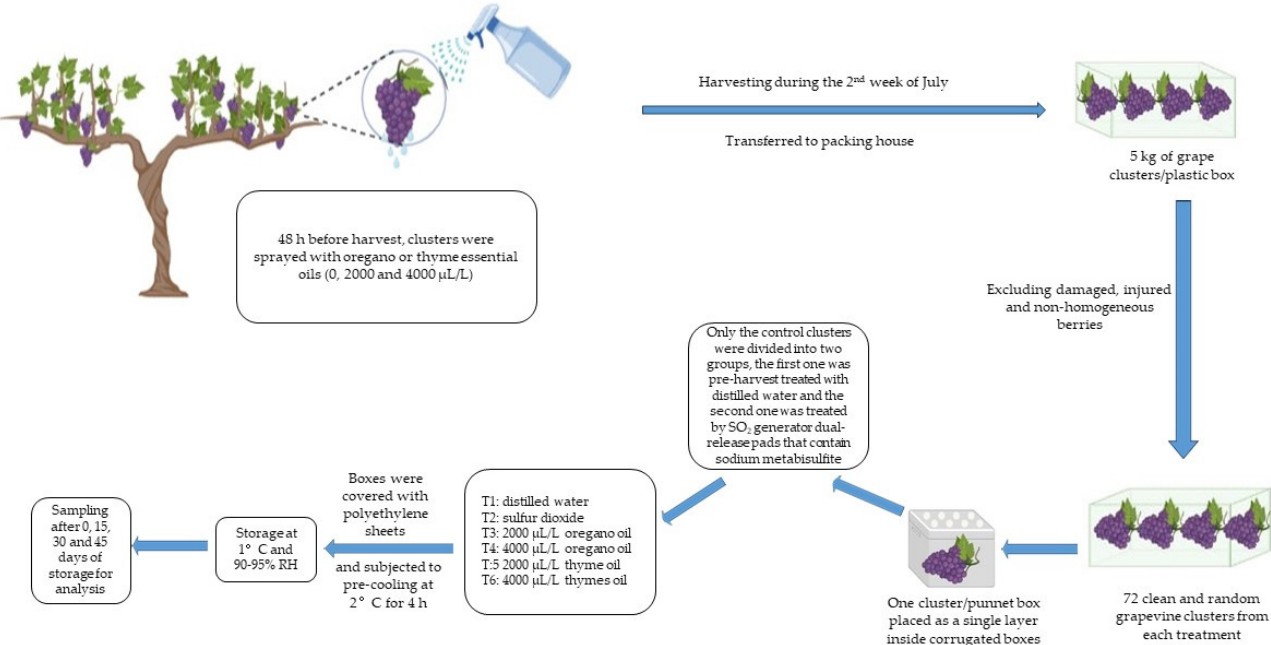

**Figure 2.** Illustration of the experimental procedures.

### 2.3. Quality Assessments

### 2.3.1. Fruit Physical Attributes

Fruit weight loss (%) was calculated every 15 days during the cold storage period using a bench-top digital scale (Model PC-500, Doran scales, Batavia, IL, USA) and calculated by the following formula: Fruit weight loss (%) = [(Fruit weight before storage − fruit weight after each cold storage period)/(Fruit weight before storage) × 100]. Berry shattering (%) was calculated according to the following equation: Berry shattering (%) = [(number of shattered berries after each cold storage period)/(total number of berries per cluster before storage) × 100]. Fruit decay caused by various fungi or microorganisms was observed every 15 days of cold storage using the following equation: Berry decay (%) = [(number of decayed berries at specified storage period)/(number of stored berries) × 100]. The incidence of naturally occurring gray mold was recorded every 15 days of cold storage, and the percentage of rotten berries was calculated according to Junior et al. [46]. Decay index due to natural infection or gray mold rot caused by *Botrytis cinerea* was assessed during the cold storage period. The average rot severity was evaluated by means of different empirical scales, based on the number of infected berries as previously described by Aghdam and Fard [47]. Disease severity scores on bunches were rated using scale ranging from 0–5 where 0 = healthy bunch without disease symptoms, 1 = ≤5% of the bunch rotten, 2 = 6–25% of

the bunch rotten, 3 = 26–50% of the bunch rotten, 4 = 51–75% of the bunch rotten, and 5 = 76 to 100% of the bunch rotten. The index was calculated by the formula: Decay index % = [Σ (n × v)/N × D) × 100], where: n = number of decayed bunches in each scale; v = degrees of rot severity scored, N = total number of examined bunches; D = the highest score of decay.

Marketable fruit (%) was calculated by the following formula: Marketable (%) = [(weight of sound fruits at specified storage period)/(Initial weight of stored fruit) × 100].

Rachis browning index (RBI) was evaluated as described by Crisosto et al. [13]. The bunches were graded on a scale of 1–4, where 1 = healthy, entire rachis including the pedicels are green and healthy, 2 = slight, rachis in good condition, but noticeable browning of pedicels, 3 = moderate, browning of pedicels and secondary rachis, 4 = severe, pedicels, secondary and primary rachis completely brown. The rachis browning index was calculated using the following formula: RBI = ∑ (browning level × number of bunches at the level)/total number of assessed bunches.

Berry firmness was measured using fruit texture Effegi penetrometer supplemented with a plunger penetrator 2 mm in diameter (FT-02, Alfonsine, Italy) at two equatorial opposite sites [48]. Two readings were taken of each berry.

The berries' visual appearance was graded for quality based on color, texture, and overall acceptability as described by Amerine et al. [49]. A nine-point hedonic scale was used (excellent = 9, very good = 8, good = 7, acceptable, moderately = 6, acceptable, slightly = 5, unacceptable, slightly = 4, unacceptable, moderately = 3, poor, very much = 2, and poor, extremely = 1).

### 2.3.2. Fruit Physio-Biochemical Attributes

The observations of berry characteristics were recorded based on fifty berries taken from 10 bunches.

### Ascorbic Acid (AsA) Content

AsA content was determined according to AOAC [50] and expressed as mg/100 mL of the juice.

Soluble solids content (SSC) was measured using a digital refractometer, (Model PAL-13810, Atago Co., Ltd., Tokyo, Japan); the results were expressed in °Brix according to AOAC [50].

Titratable acidity (TA%) was assayed as described by AOAC [50]. An aliquot of fruit juice was taken and titrated against 0.1 N NaOH in the presence of phenolphthalein as an indicator to the end point and was calculated as grams of tartaric acid per 100 mL of juice.

SSC/TA ratio was calculated from the values recorded for fruit juice SSC and TA percentages were determined.

Total anthocyanin content (TAC) in berry skin was determined using methanolic HCl extraction as described by Fuleki and Francis [51]. Briefly, a peel sample (0.5 g) was randomly collected, mixed with acidic methanol (30 mL), and left in dark conditions at room temperature (≈20–22 °C) for 48 h. The absorbance was measured at 535 nm wavelength by spectrophotometer (Model RT2, ThermoFisher Scientific, Waltham, MA, USA) and results were expressed as mg $100 \, g^{-1}$ of berry skin.

### 2.3.3. Enzyme Activities
### Peroxidase (POX) Activity (EC 1.11.1.7)

The POX activity was assayed using guaiacol as a donor and $H_2O_2$ as a substrate according to the method of Herzog and Fahimi [52]. One gram of berry sample was homogenized in 10 mL of 0.1 M sodium phosphate buffer solution (pH = 6.5) and the homogenate was centrifuged at 20,000× *g* at 4 °C for 15 min. The assay mixture (100 mL) contained 10 mL of 1% (*v/v*) guaiacol, 10 mL of 0.3% $H_2O_2$, and 80 mL of 50 mM sodium phosphate buffer solution (pH 6.4). A quantity of 200 μL of supernatant was added to 2.8 mL of the assay mixture to start the reaction. The increase in absorbance was recorded

every 30 s for 3 min at 460 nm by using a spectrophotometer (Model RT2, ThermoFisher Scientific, Waltham, MA, USA). One unit of POX activity was defined as the amount of enzyme that caused an increase in absorbance of optical density at 460 nm per min under the assay conditions and the results were expressed as U min$^{-1}$ g$^{-1}$ fresh weight.

Polyphenol Oxidase (PPO) Activity (EC 1.14.18.1)

The activity of PPO was assayed according to Mayer et al. [53]. Briefly, one gram of berry sample was homogenized in 10 mL of 0.1 M sodium phosphate buffer solution (pH 6.4) and the homogenate was centrifuged at 20,000× *g* at 4 °C for 15 min. The supernatant was used to determine PPO activity with catechol as substrate. The reaction mixture consisted of 200 μL of the extract, 1.5 mL of 0.1 M sodium phosphate buffer solution, and 200 μL of 0.1 M catechol. The activity was expressed as change in absorbance at 495 nm and measured using a spectrophotometer Model RT2 (ThermoFisher Scientific, Waltham, MA, USA) at 30 s intervals for 3 min, and the results were represented in U min$^{-1}$ g$^{-1}$ fresh weight.

Pectin Methylesterase (PME) Activity (EC 3.1.1.11)

The PME was extracted and assayed by the method of Hagerman and Austin [54] with some modification. The activity was assayed in a mixture containing 2 mL of citrus pectin solution (0.01%), 0.2 mL NaCl (0.15 M), 0.1 mL of bromothymol blue solution (0.01%), 0.6 mL of distilled water, and 0.1 mL of enzymatic extract. All solutions (pectin, indicator dye, and distilled water) were adjusted to pH 7.5 using 0.1 N NaOH just before starting each trial. After adding the prepared enzyme, the cuvette was gently shaken. The absorbance was measured immediately at 620 nm by using a spectrophotometer (Model RT2, ThermoFisher Scientific, Waltham, MA, USA) and measured again after 3 min. The different values in absorbance between 0 and 3 min were the measure of PME activity. One unit was defined as the amount of enzyme required for liberating one μmol of methyl ester and the results were expressed as U min$^{-1}$ g$^{-1}$ fresh weight.

### 2.3.4. Shelf-Life

The shelf-life of grape clusters was determined by observing the number of days that the clusters remained acceptable for marketing according to Mondal [55].

### 2.3.5. Economic Viability

The economic viability of each treatment was calculated using the material price and the quantity required per ton of grape clusters.

### 2.4. Experimental Design and Statistical Data Analysis

This experiment was arranged as a randomized complete block design with three replicates. Data were preliminarily tested for numerical normality and homogeneity of variance using the Shapiro–Wilk and Levene tests, respectively. Data calculated as percentages were first transformed to the Arcsine square root values before performing the analysis of variance (ANOVA), and results were presented as back-transformed means. The ANOVA was performed using the CoStat software package (version 6.311, CoHort software, Monterey, CA, USA). Mean comparisons were conducted, where appropriate, using Tukey's honestly significant difference (HSD) test at probability (*p*) $\leq 0.05$ [56]. The fruit characteristics were generated using a principal component analysis (PCA) [57]. The PCA was performed using JMP Pro 16 (SAS Institute, Cary, NC, USA). The SO$_2$ treatment is not displayed in physio-biochemical attributes and enzyme activities because this treatment was started with the beginning of storage.

### 3. Results

*3.1. Effect of Preharvest Applications of Oregano and Thyme Essential Oils on Physical Properties of 'Flame Seedless' Grapes during 45 Days of Cold Storage*

3.1.1. Fruit Physical Attributes

Fruit weight loss increased with the length of the storage period under all treatments (Figure 3). The lowest weight loss was found under preharvest treatment with oregano essential oil at 2000 µL/L in both seasons, while the highest weight loss was recorded in clusters sprayed with distilled water (control).

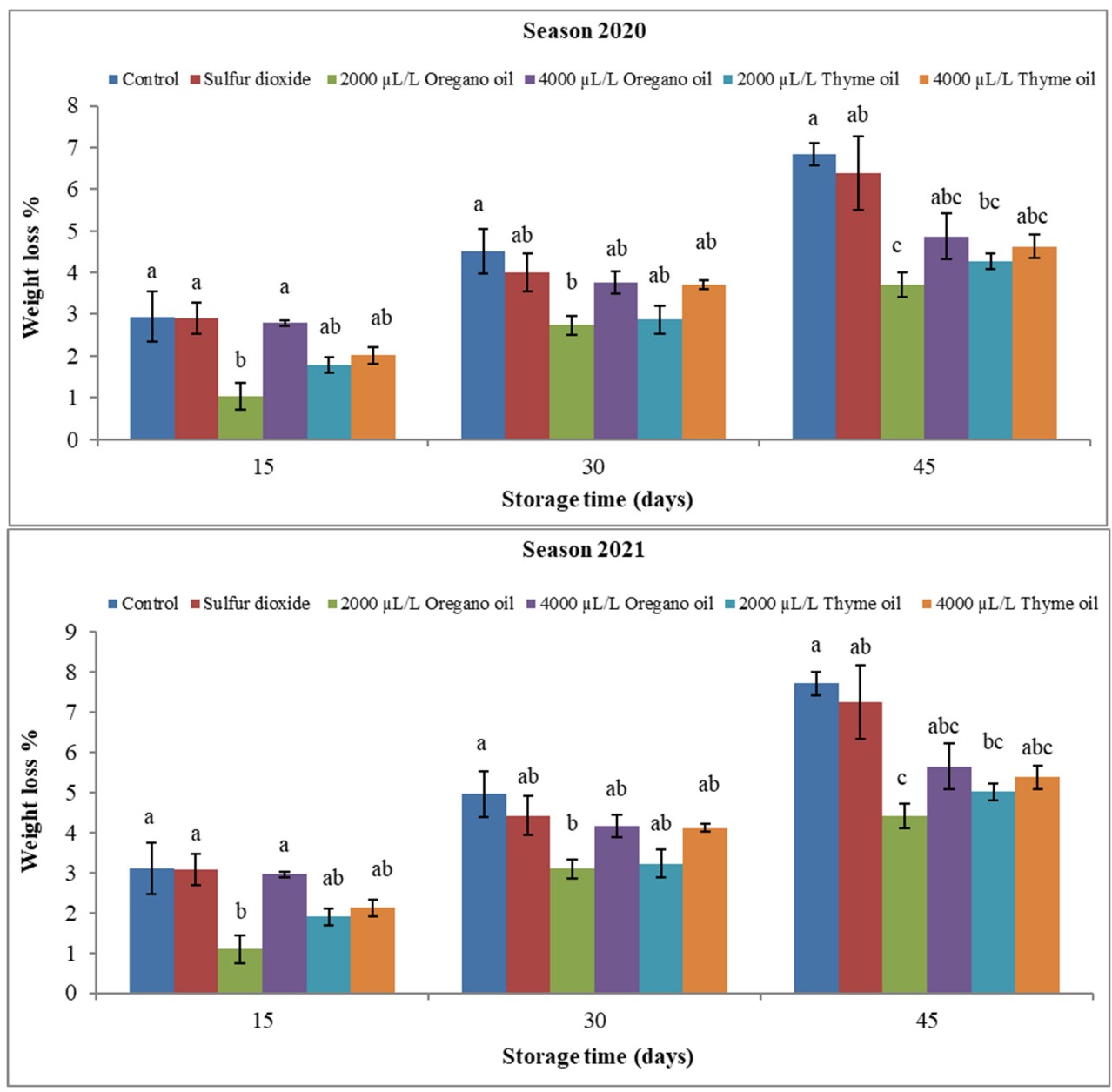

**Figure 3.** Effect of preharvest applications of oregano and thyme EOs on weight loss (%) of 'Flame Seedless' grapes during cold storage at 1 °C. Each value represents the mean ± SE of three replicates. Means followed by the same letters in each season and storage period are not significantly different according to Tukey's HSD test at $p \leq 0.05$.

The decay incidence significantly decreased due to sulfur dioxide and EO treatments at all storage periods compared to the control (nontreated bunches) which showed the highest percentage of decay (Figure 4). In the first season, no significant difference was noted among sulfur dioxide and EO treatments until 30 days of storage, while after 45 days, EO treatments exhibited a significant reduction in decay percentage. The effectiveness of EO treatments was more pronounced in the second season starting from 30 days of storage compared to control and sulfur dioxide treatments. The lowest decay percentage was found with oregano essential oil treatment at 2000 µL/L.

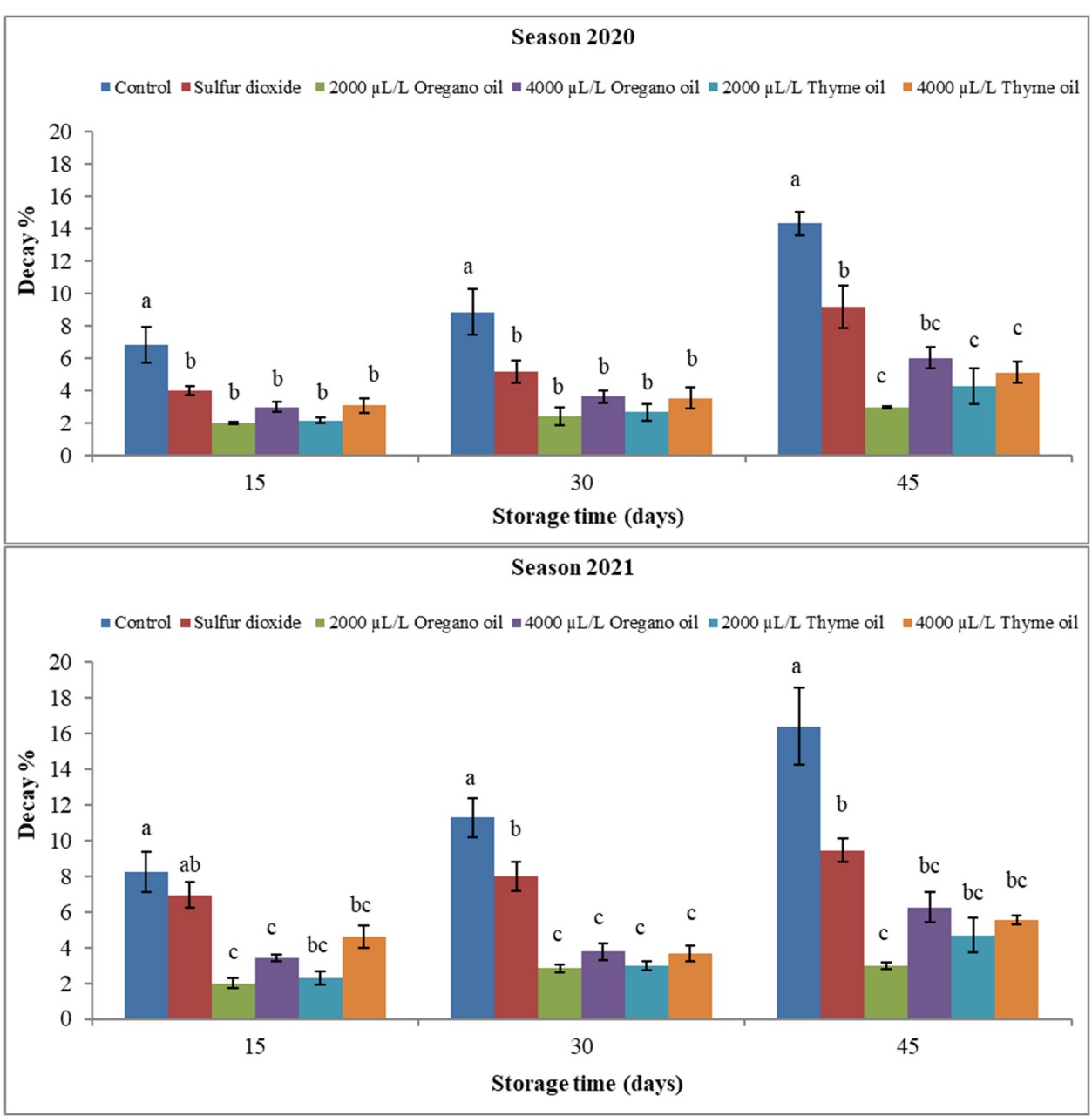

**Figure 4.** Effect of preharvest applications of oregano and thyme EOs on decay incidence (%) in 'Flame Seedless' grapes during cold storage at 1 °C. Each value represents the mean ± SE of three replicates. Means followed by the same letters in each season and storage period are not significantly different according to Tukey's HSD test at $p \leq 0.05$.

The gray mold percentage followed the same pattern as the decay percentage (Figure 5). Preharvest treatments with oregano and thyme EOs at 2000 µL/L had the best consistent results in reducing gray mold incidence in the two seasons, especially at the end of the storage period. This was true also for the decay index (Figure 6), indicating the effectiveness of both EOs used in reducing decay and gray mold incidence compared to the control and sulfur dioxide.

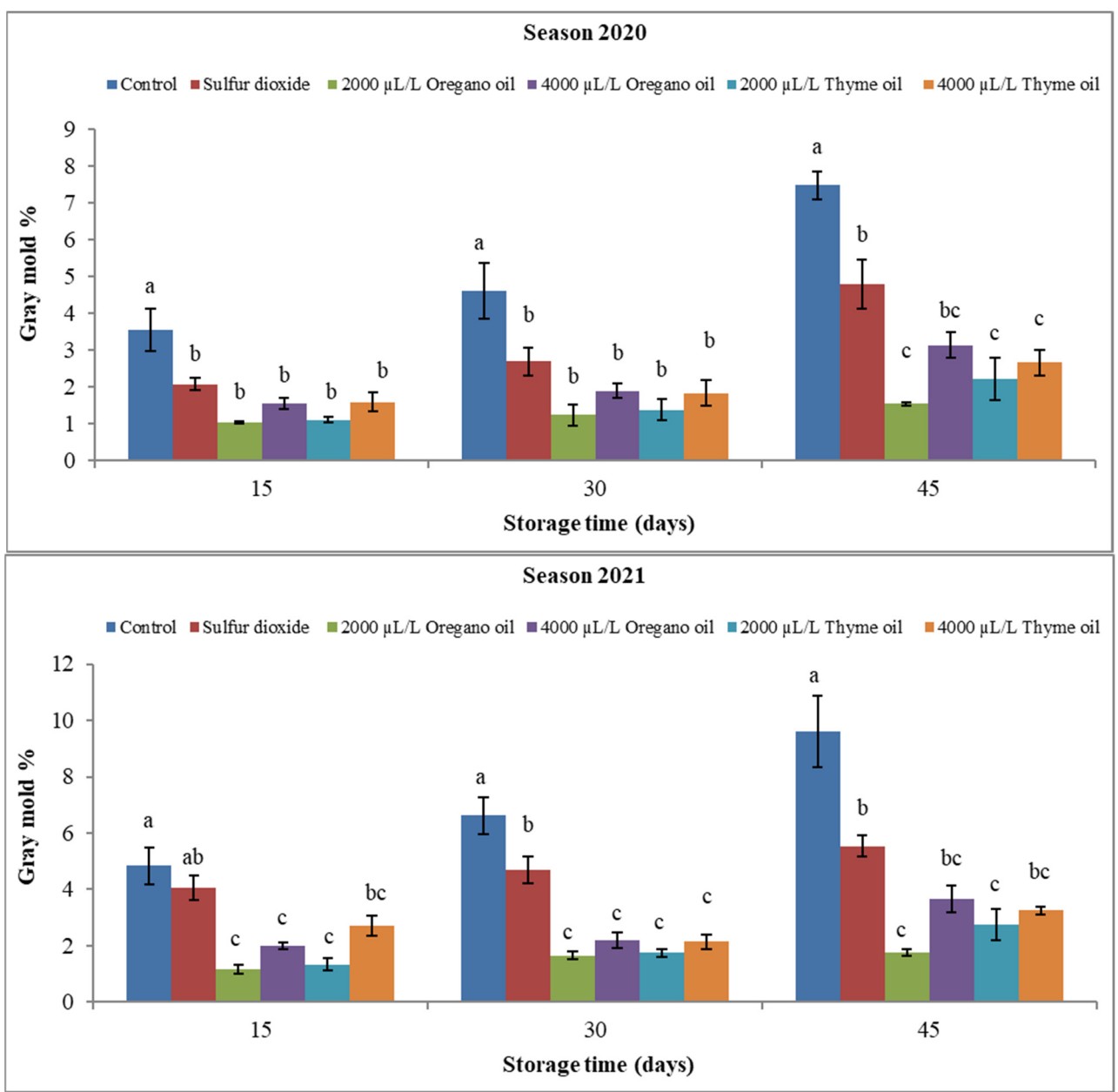

**Figure 5.** Effect of preharvest applications of oregano and thyme EOs on gray mold (%) in 'Flame Seedless' grapes during cold storage at 1 °C. Each value represents the mean ± SE of three replicates. Means followed by the same letters in each season and storage period are not significantly different according to Tukey's HSD test at $p \leq 0.05$.

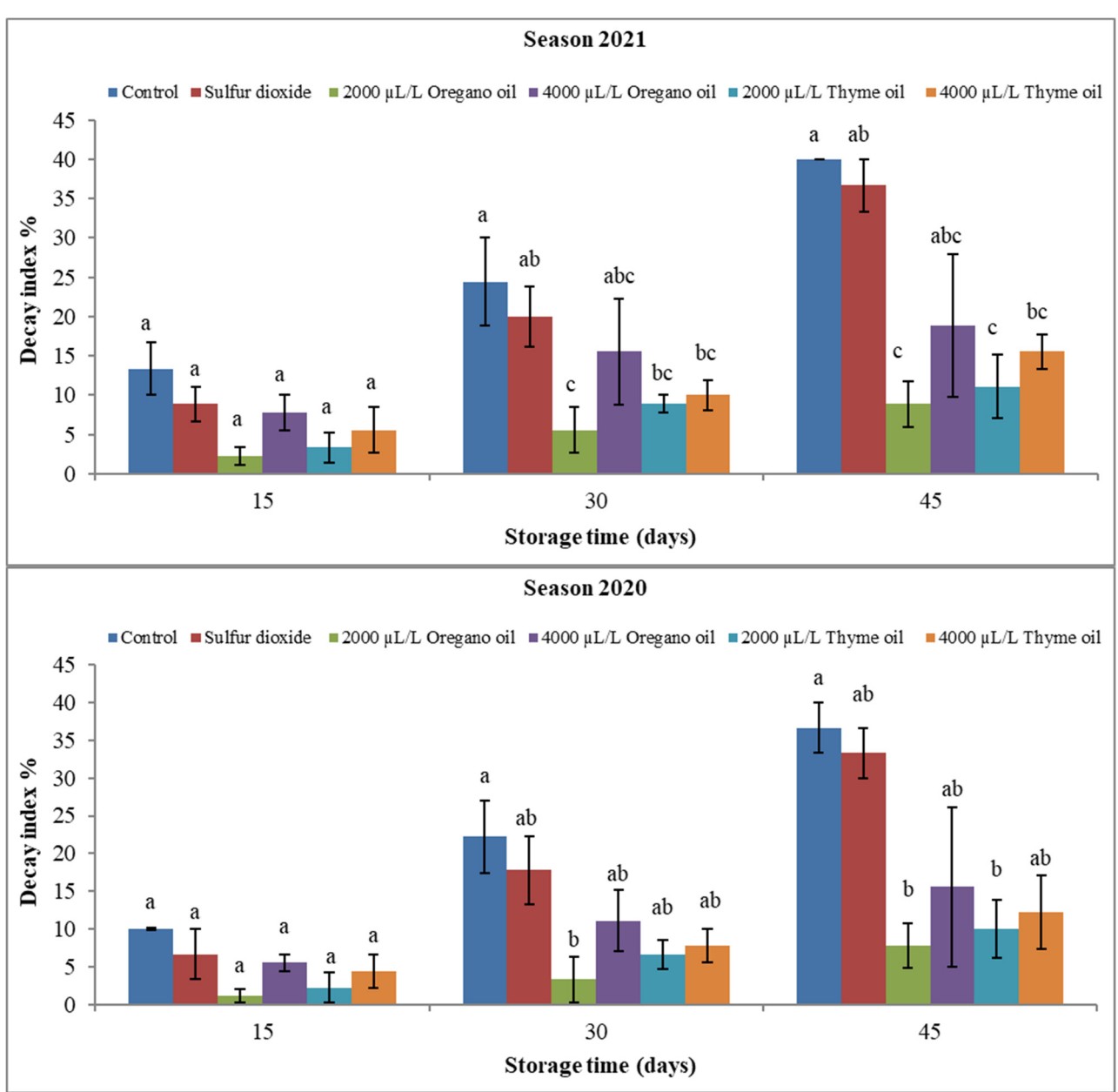

**Figure 6.** Effect of preharvest applications of oregano and thyme EOs on decay index in 'Flame Seedless' grapes during cold storage at 1 °C. Each value represents the mean ± SE of three replicates. Means followed by the same letters in each season and storage period are not significantly different according to Tukey's HSD test at $p \leq 0.05$.

3.1.2. Marketable Fruit Percentage

The marketable fruit percentage decreased when extending the cold storage period, and recorded the lowest values after 45 days of storage (Figure 7). Oregano and thyme EO treatments generally maintained higher marketable percentages throughout the storage period compared to control and sulfur dioxide, respectively. After 45 days of cold storage, the lowest values were from the nontreated control followed by the sulfur dioxide treatment. Moreover, the highest marketable fruit percentage was obtained when oregano EO at 2000 µL/L was applied prior to harvest, but did not differ from marketable fruit percentage when thyme EO at 2000 and 4000 µL/L were applied prior to harvest.

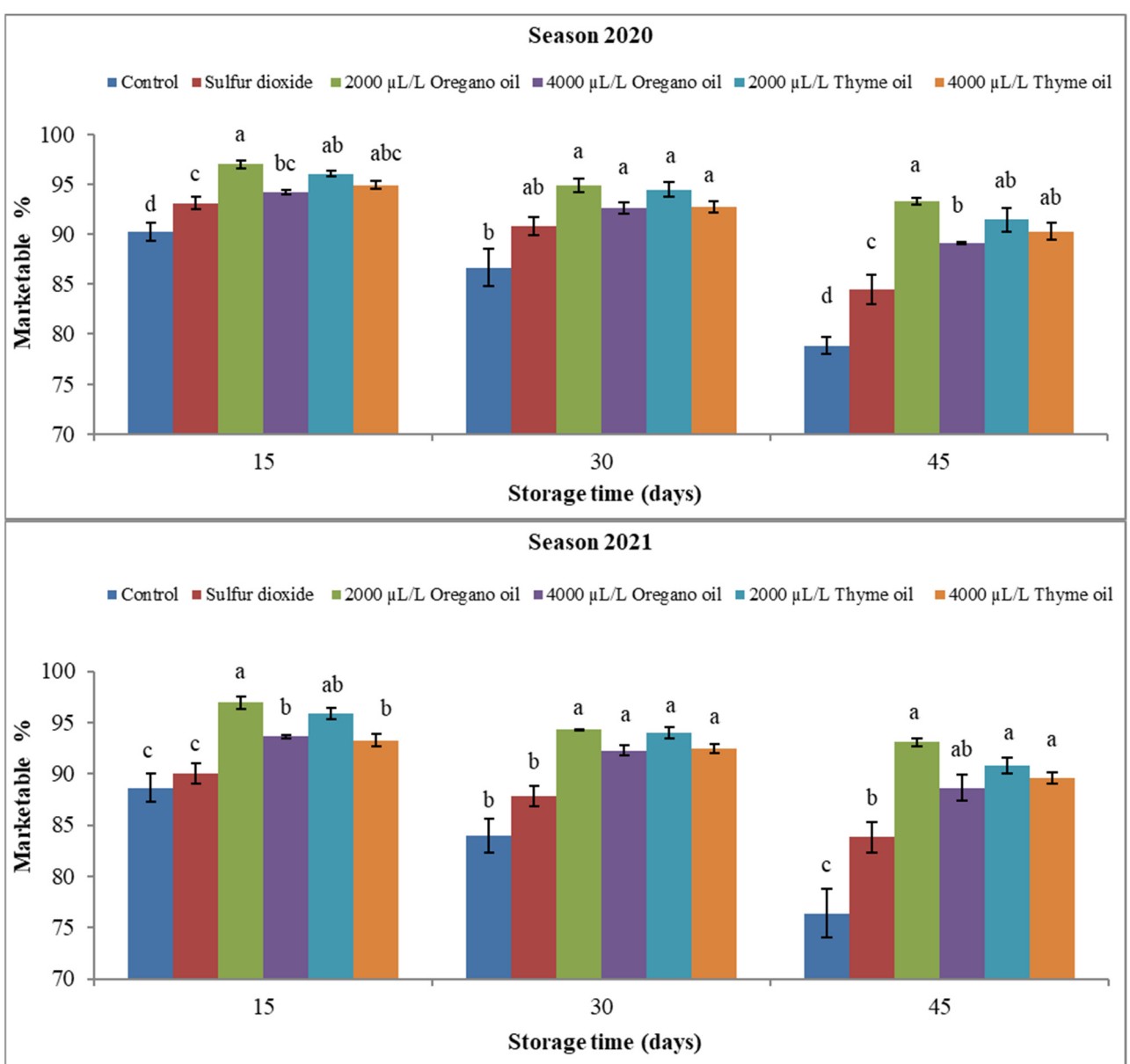

**Figure 7.** Effect of preharvest applications of oregano and thyme EOs on marketable fruit percentage of 'Flame Seedless' grapes during cold storage at 1 °C. Each value represents the mean ± SE of three replicates. Means followed by the same letters in each season and storage period are not significantly different according to Tukey's HSD test at $p \leq 0.05$.

### 3.1.3. Rachis Browning Index (RBI) and Berry Shattering (BS) Percentage

Oregano and thyme EO treatments significantly reduced the RBI across all storage periods when compared to the control and sulfur dioxide treatments (Figure 8). There were no significant differences among oregano and thyme EO treatments, particularly after 15 and 45 days of storage.

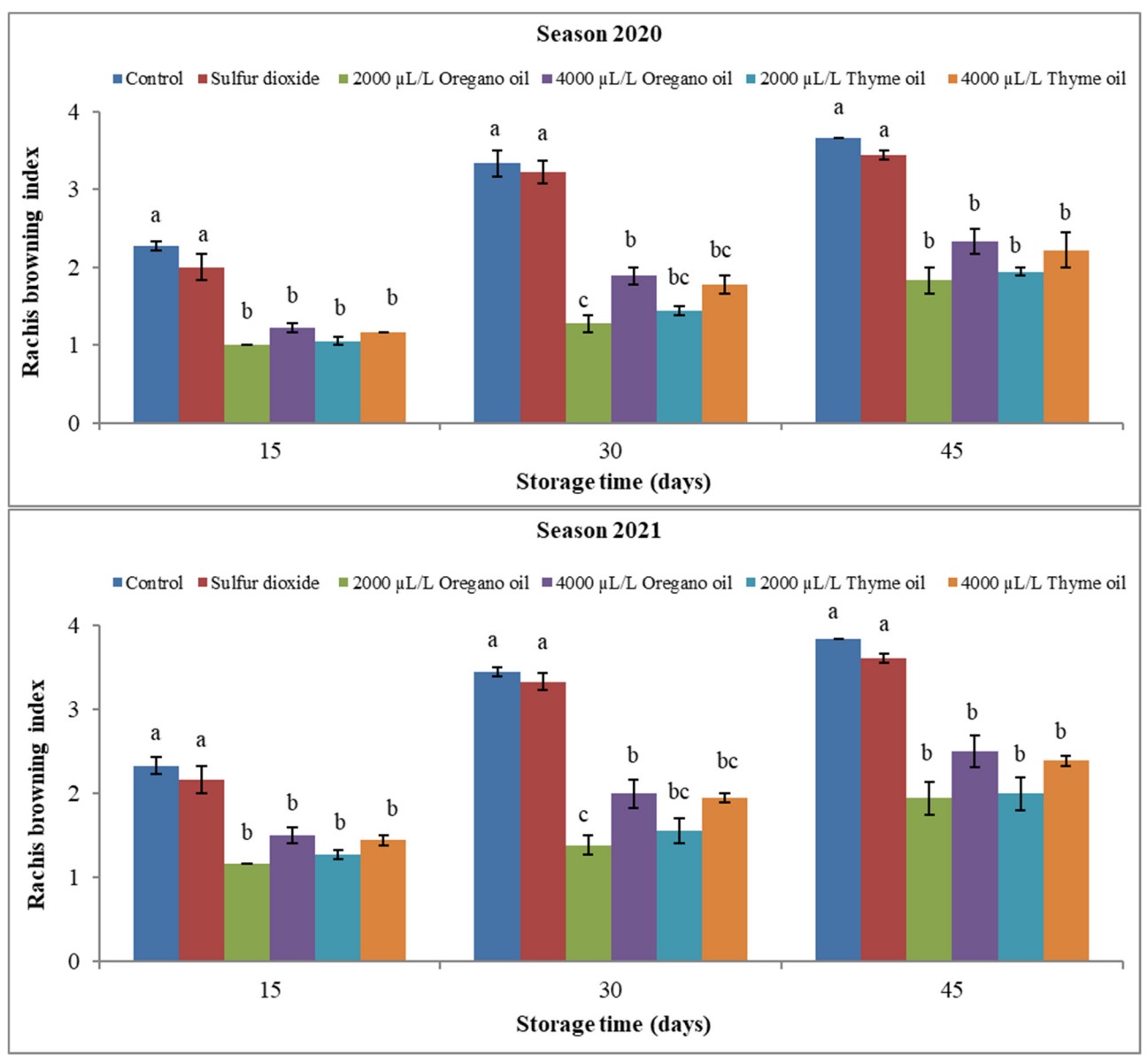

**Figure 8.** Effect of preharvest applications of oregano and thyme EOs on the rachis browning index of 'Flame Seedless' grapes during cold storage at 1 °C. Each value represents the mean ± SE of three replicates. Means followed by the same letters in each season and storage period are not significantly different according to Tukey's HSD test at $p \leq 0.05$.

Berry shattering (BS) increased with prolonging storage period in both seasons (Figure 9). No significant difference among treatments was noted after 15 days of cold storage; however, the control had the highest BS and the lowest was at oregano and thyme EO treatments at 2000 µL/L, especially in the second season after 45 days of storage.

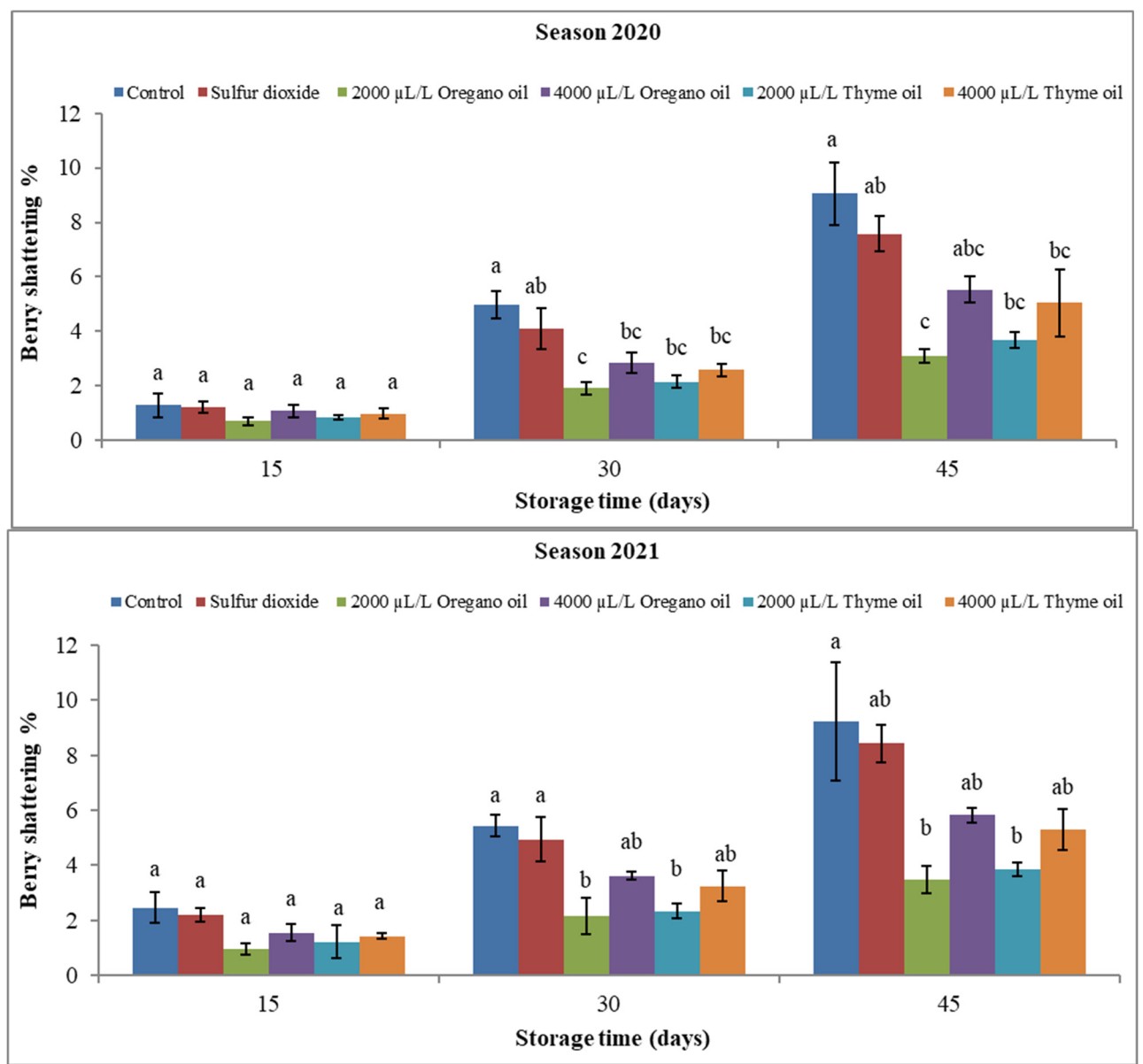

**Figure 9.** Effect of preharvest applications of oregano and thyme EOs on berry shattering percentage of 'Flame Seedless' grapes during cold storage at 1 °C. Each value represents the mean ± SE of three replicates. Means followed by the same letters in each season and storage period are not significantly different according to Tukey's HSD test at $p \leq 0.05$.

### 3.1.4. Berry Visual Appearance Score and Firmness

The visual appearance of 'Flame Seedless' grape clusters was decreased when extending the cold storage period in both seasons, where the lowest visual appearance scores were obtained after 45 days of storage under all treatments (Figure 10). Oregano and thyme EO treatments at 2000 µL/L had the highest score compared to the control and sulfur dioxide, particularly in the second season after 45 days of storage.

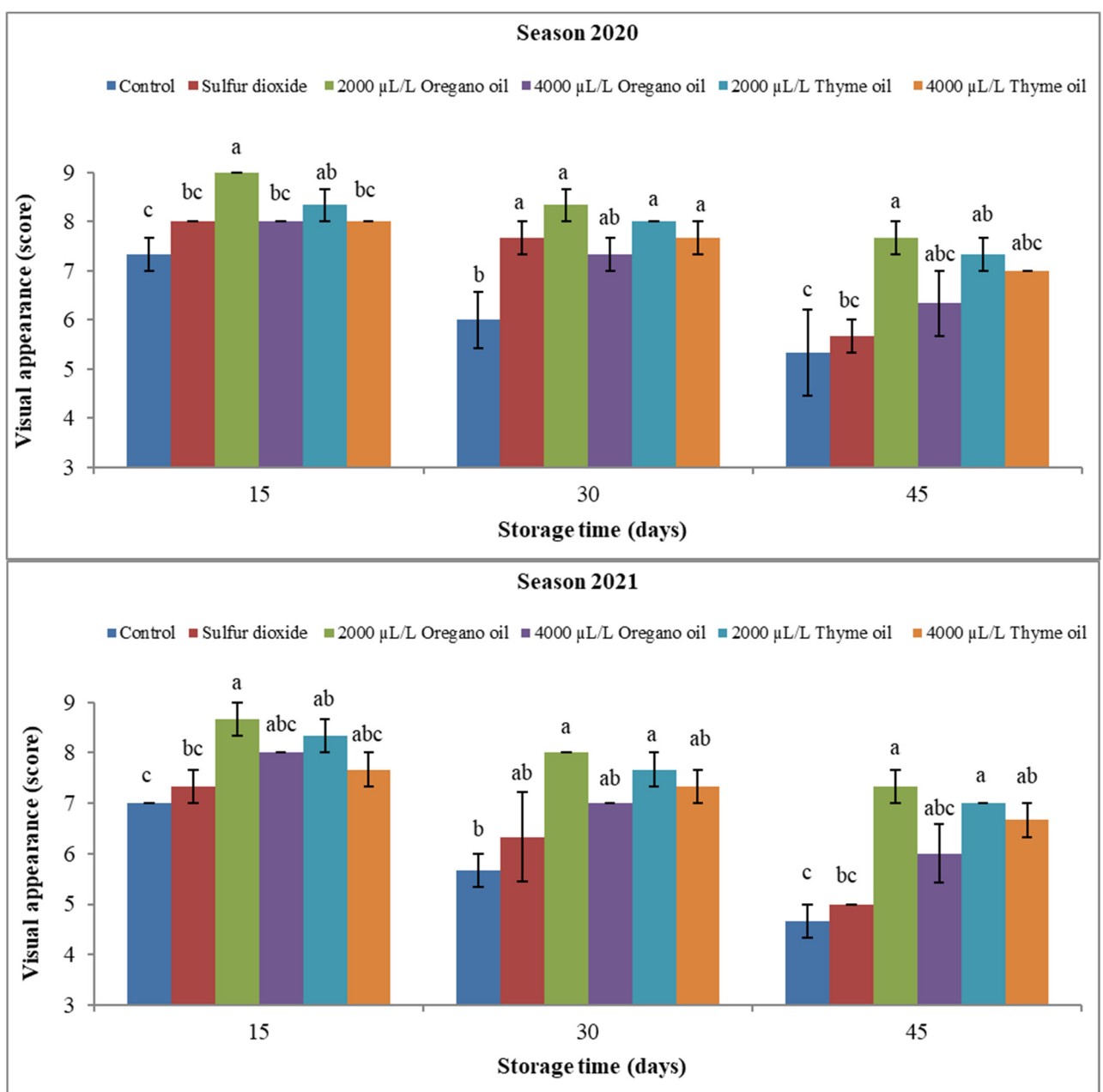

**Figure 10.** Effect of preharvest applications of oregano and thyme EOs on visual appearance score of 'Flame Seedless' grapes during cold storage at 1 °C. Each value represents the mean ± SE of three replicates. Means followed by the same letters in each season and storage period are not significantly different according to Tukey's HSD test at $p \leq 0.05$.

The lowest berry firmness was found with the nontreated (control) in both seasons under all storage periods (Figure 11). After 45 days, there was no significant difference between sulfur dioxide and EO treatments, especially in the first season. However, in the second season, the highest firmness was achieved with oregano and thyme EO treatments at 2000 µL/L.

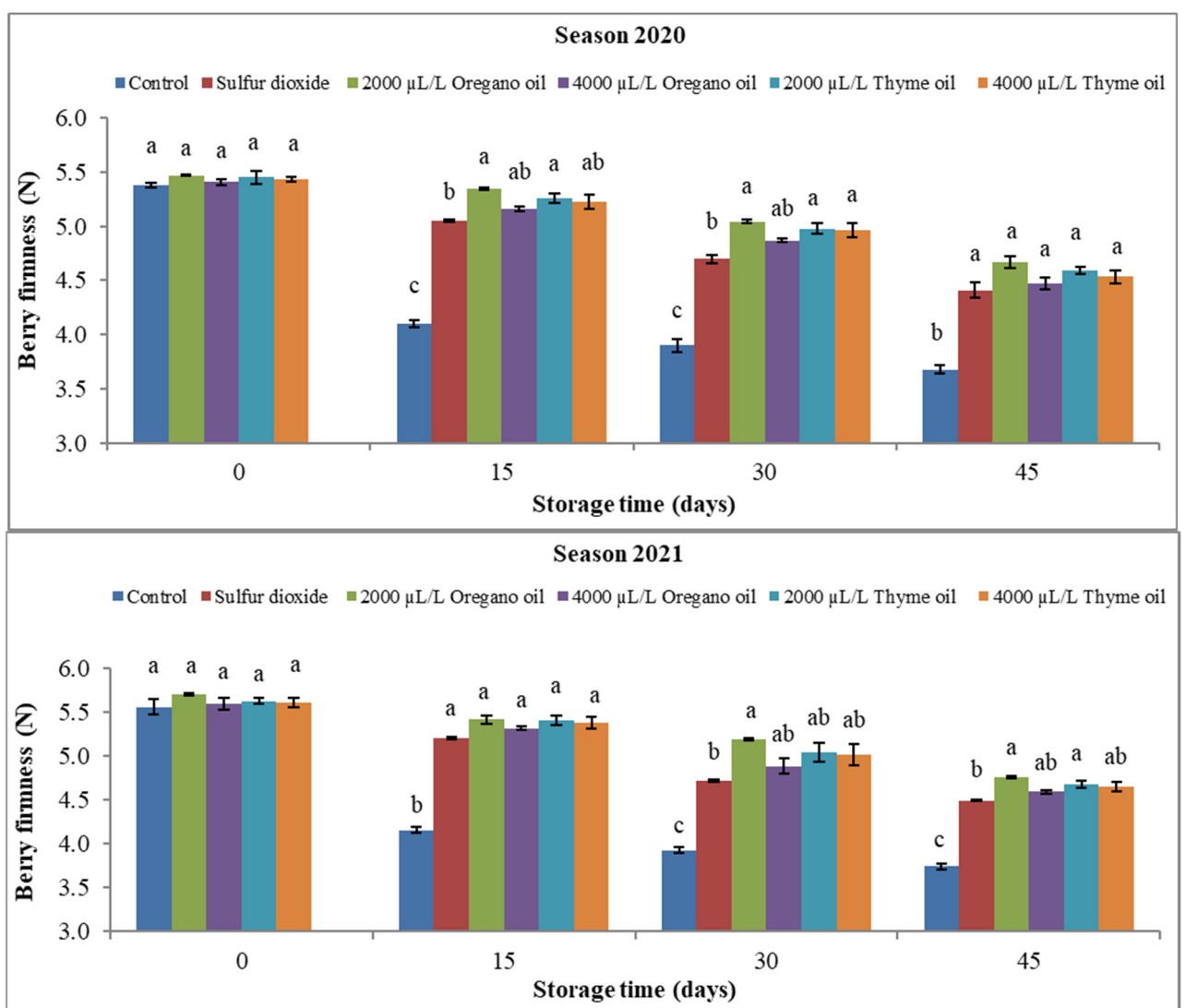

**Figure 11.** Effect of preharvest applications of oregano and thyme EOs on berry firmness (N) of 'Flame Seedless' grapes during cold storage at 1 °C. Each value represents the mean ± SE of three replicates. Means followed by the same letters in each season and storage period are not significantly different according to Tukey's HSD test at $p \leq 0.05$.

*3.2. Effect of Preharvest Applications of Oregano and Thyme Essential Oils on Physio-Biochemical Properties of 'Flame Seedless' Grapes during 45 Days of Cold Storage*

3.2.1. Berry Ascorbic Acid (AsA) Content

The AsA content decreased from harvest to the end of the storage period (Figure 12). No significant differences were found among all treatments at harvest and after 15 days of storage. However, oregano EO treatment at 2000 µL/L had the highest AsA content after 30 and 45 days of storage in both seasons but did not differ from sulfur dioxide and other EO treatments after 30 days and with the other EO treatments after 45 days. The control treatment showed the lowest content with no significant difference with sulfur dioxide treatment after 45 days.

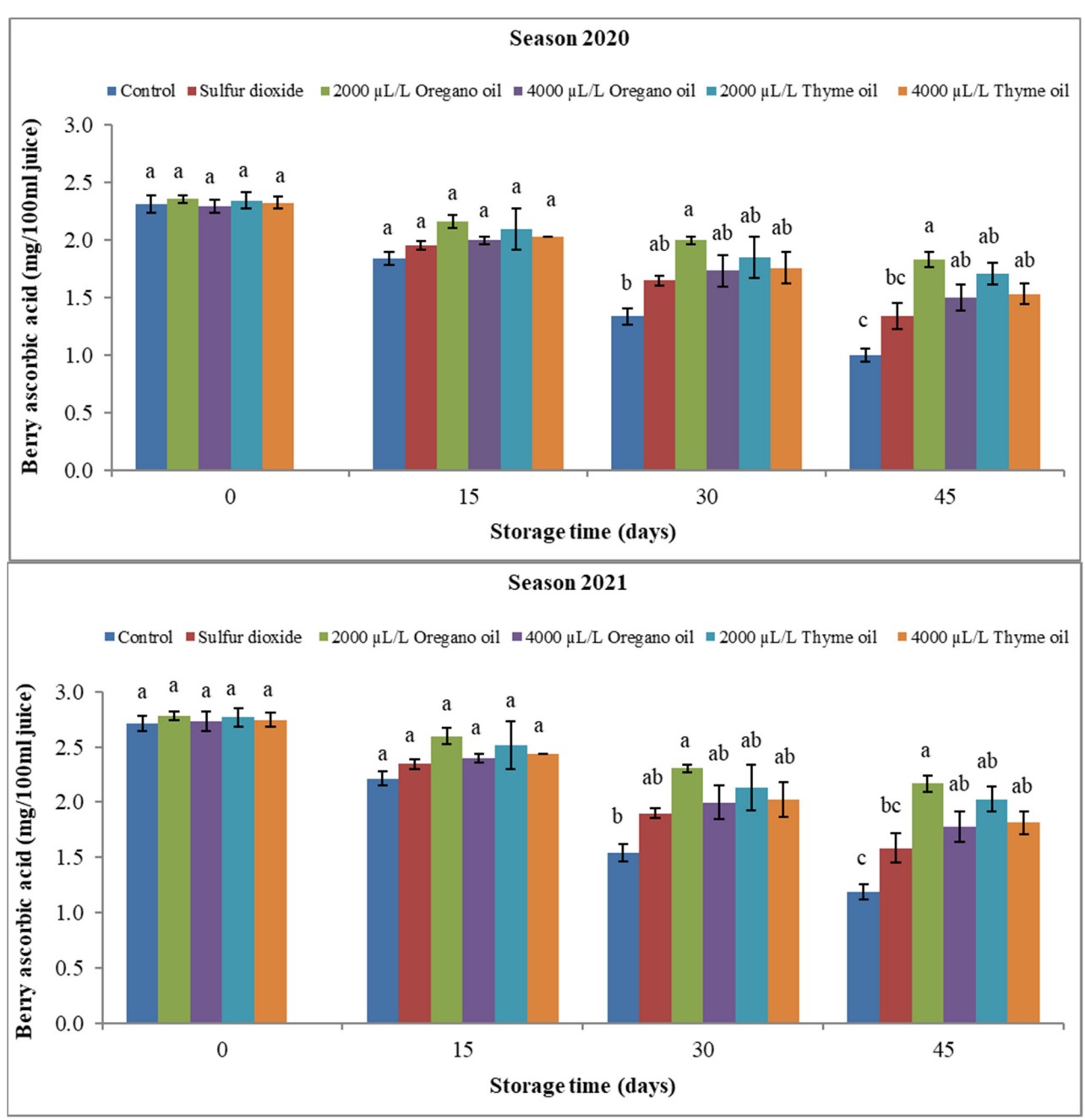

**Figure 12.** Effect of preharvest applications of oregano and thyme EOs on the berry ascorbic acid content of 'Flame Seedless' grapes during cold storage at 1 °C. Each value represents the mean ± SE of three replicates. Means followed by the same letters in each season and storage period are not significantly different according to Tukey's HSD test at $p \leq 0.05$.

### 3.2.2. Berry Soluble Solids Content (SSC), Titratable Acidity (TA), and SSC/TA

The SSC in grape berries increased gradually with the progress of storage period (Figure 13), while TA showed an opposite trend in both seasons (Figure 14). The SSC/TA ratio increased when extending the cold storage period in both seasons (Figure 15). Oregano and thyme EO treatments showed lower SSC than the control and sulfur dioxide, es-pecially after 30 and 45 days of storage with no significant difference among EO treatments and between control and sulfur dioxide treatment (Figure 13).

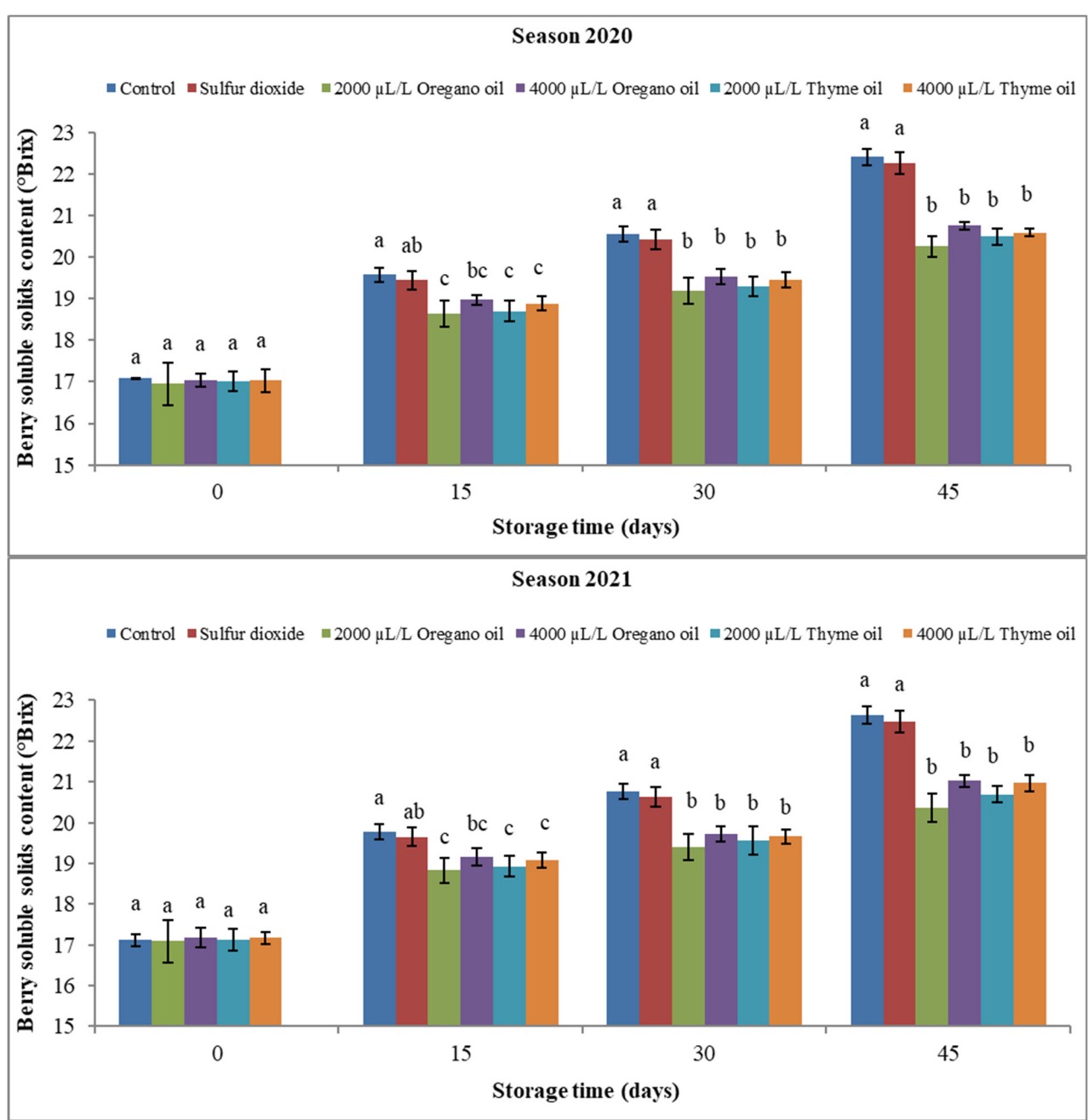

**Figure 13.** Effect of preharvest applications of oregano and thyme EOs on the berry soluble solids content of 'Flame Seedless' grapes during cold storage at 1 °C. Each value represents the mean ± SE of three replicates. Means followed by the same letters in each season and storage period are not significantly different according to Tukey's HSD test at $p \leq 0.05$.

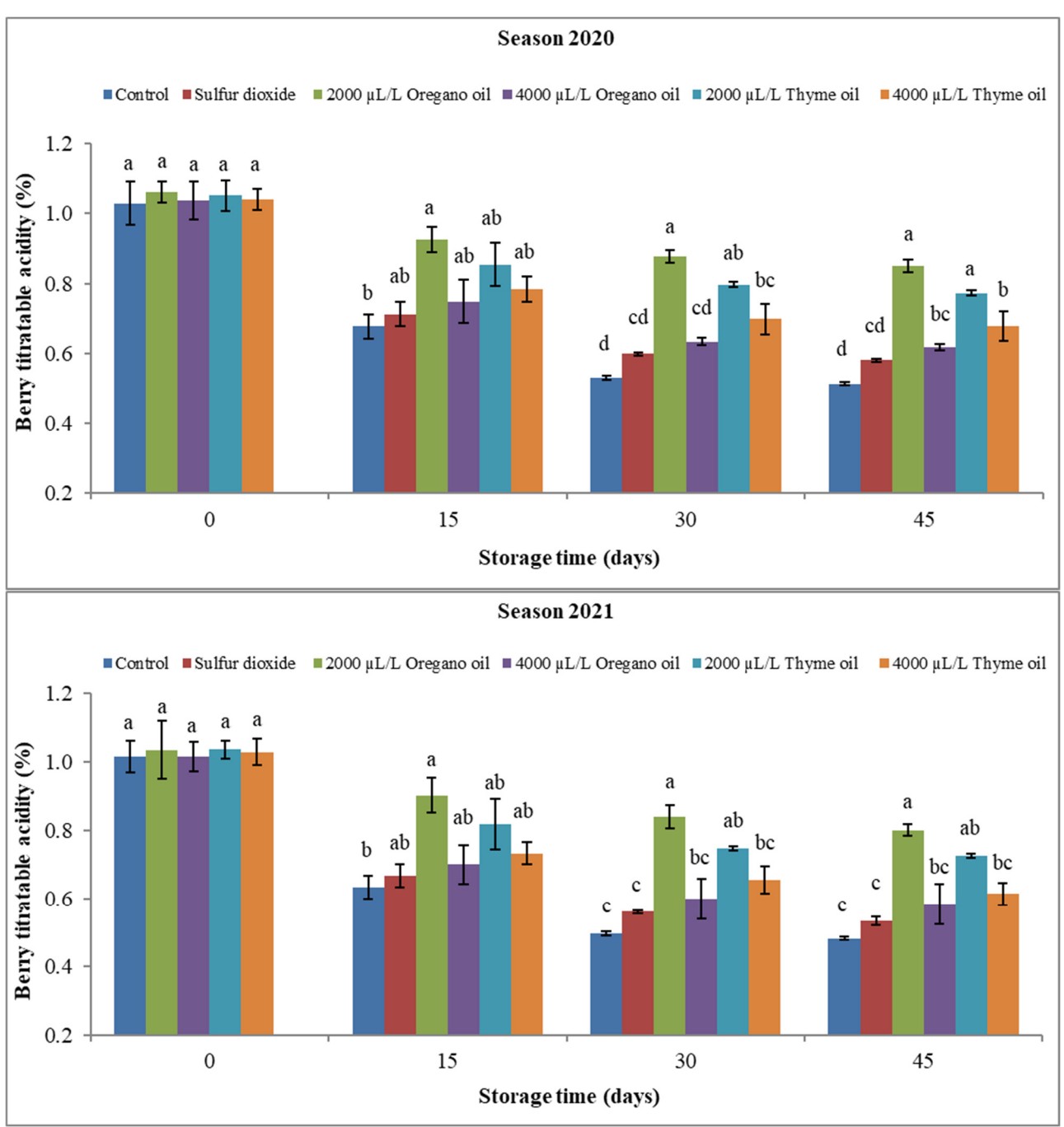

**Figure 14.** Effect of preharvest applications of oregano and thyme EOs on berry titratable acidity of 'Flame Seedless' grapes during cold storage at 1 °C. Each value represents the mean ± SE of three replicates. Means followed by the same letters in each season and storage period are not significantly different according to Tukey's HSD test at $p \leq 0.05$.

The lowest TA was found under control and sulfur dioxide treatments in all storage periods, while the highest content was found under oregano and thyme EO treatments at 2000 and 4000 µL/L, respectively, especially after 30 and 45 days of storage (Figure 14). The SSC/TA ratio after 30 and 45 days showed higher values under control followed by sulfur dioxide treatment and the lowest ratio was found under oregano EO treatment at 2000 µL/L (Figure 15).

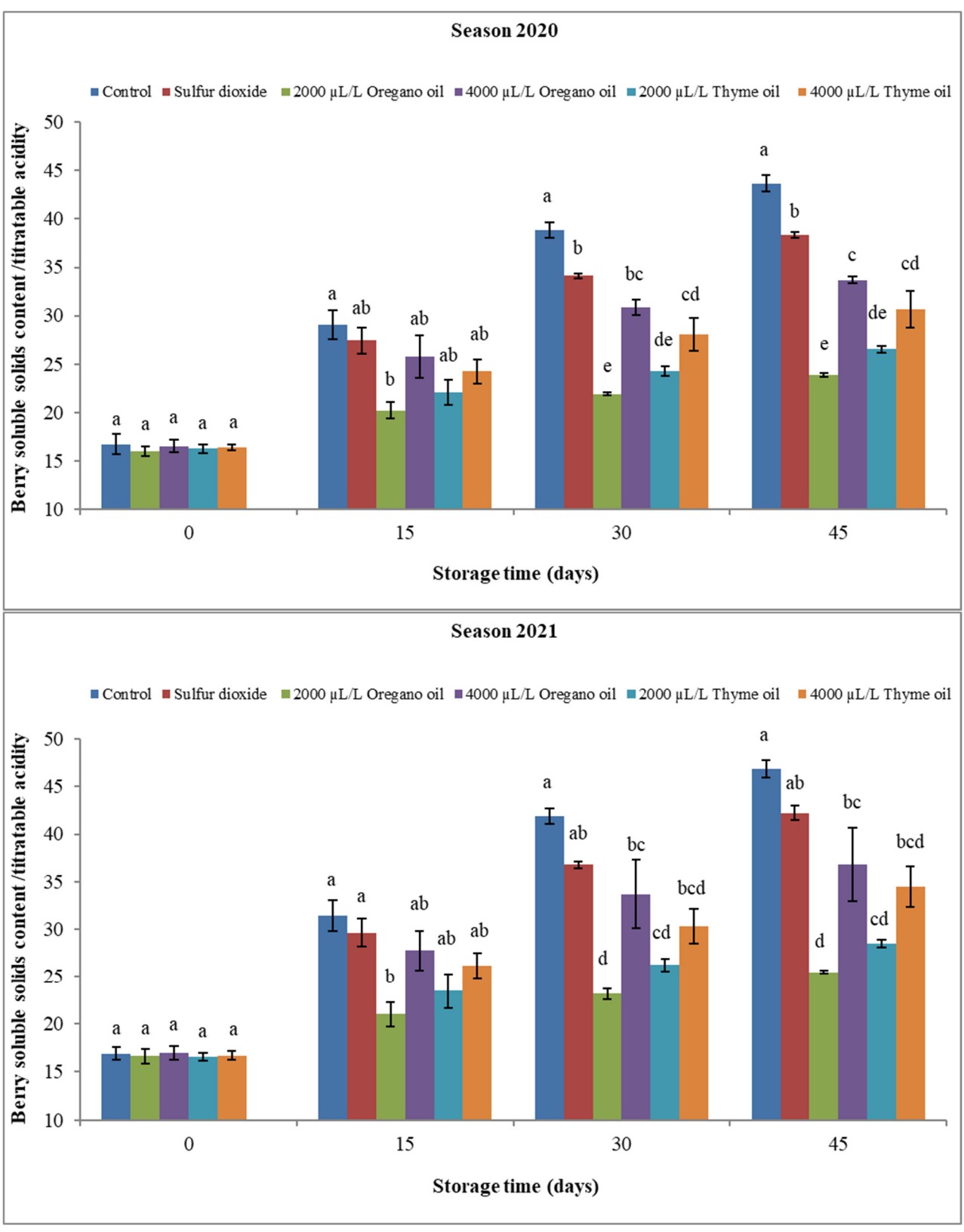

**Figure 15.** Effect of preharvest applications of oregano and thyme EOs on the berry soluble solids content/titratable acidity of 'Flame Seedless' grapes during cold storage at 1 °C. Each value represents the mean ± SE of three replicates. Means followed by the same letters in each season and storage period are not significantly different according to Tukey's HSD test at $p \leq 0.05$.

### 3.2.3. Berry Total Anthocyanin Content

Results show that total anthocyanin content increased under control and sulfur dioxide treatments after 15 days of storage, then decreased after 30 and 45 days in cold storage; however, oregano and thyme EO treatments increased berry anthocyanin content during storage periods (Figure 16). Significant differences among treatments were found at 45 days of storage in the first season, while this was found starting from 30 days in the second season. During these two specific periods, the highest contents were found with the EO treatments, while the lowest anthocyanin contents were recorded under control and sulfur dioxide treatments in both seasons.

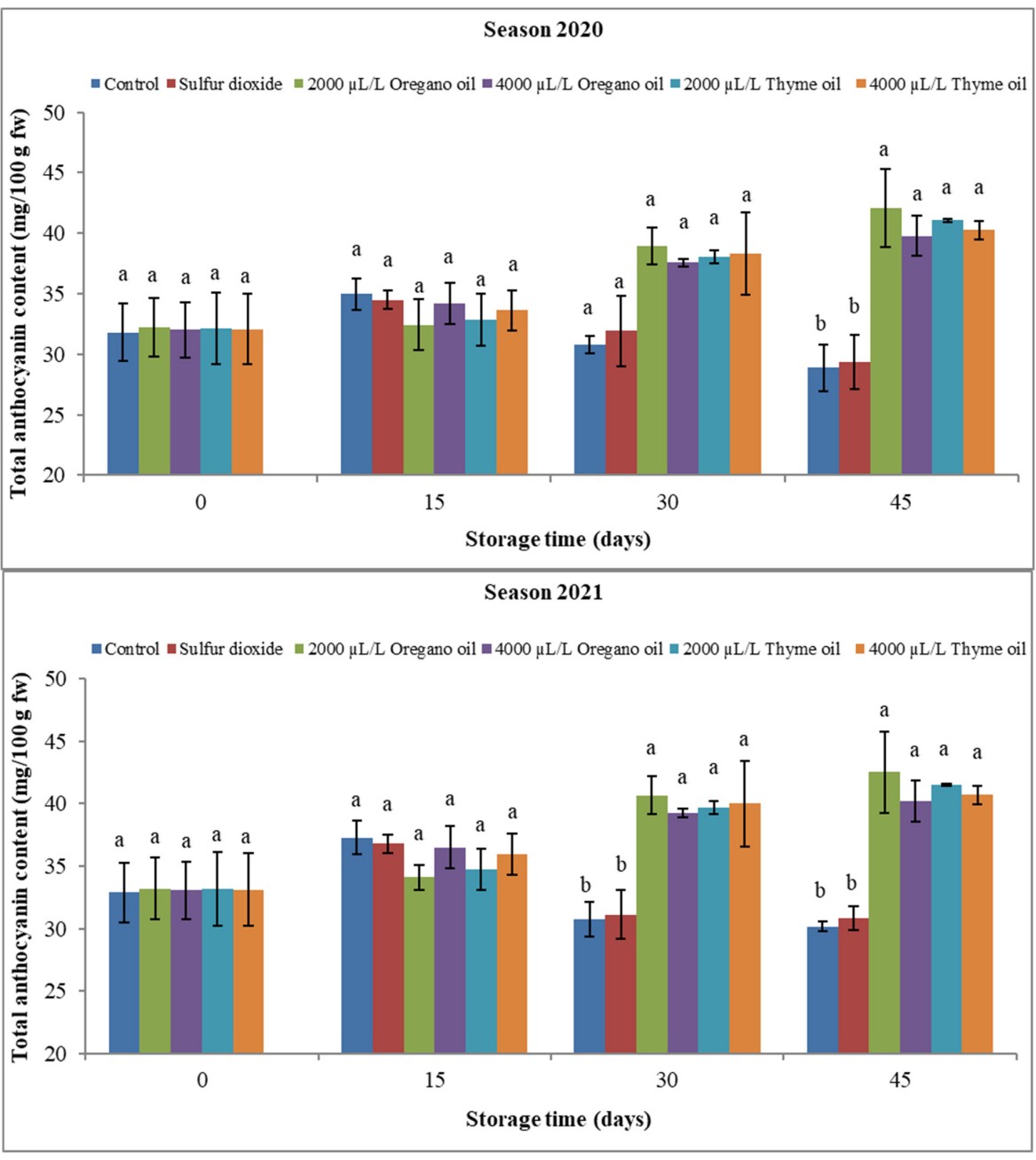

**Figure 16.** Effect of preharvest applications of oregano and thyme EOs on total anthocyanin content of 'Flame Seedless' grapes during cold storage at 1 °C. Each value represents the mean ± SE of three replicates. Means followed by the same letters in each season and storage period are not significantly different according to Tukey's HSD test at $p \leq 0.05$.

### 3.3. Effect of Preharvest Applications of Oregano and Thyme Essential Oils on Enzyme Activity of 'Flame Seedless' Grapes during 45 Days of Cold Storage

#### 3.3.1. Peroxidase (POX) and Polyphenol Oxidase (PPO)

Activities of POX and PPO enzymes were significantly affected by various applied treatments showing an increase during storage (Figures 17 and 18). However, the lowest activities were noted at 2000 µL/L with the oregano EO treatment followed by thyme EO treatment at the same concentration, then oregano and thyme EO treatments at 4000 µL/L, and sulfur dioxide treatment, respectively, after 30 and 45 days of storage.

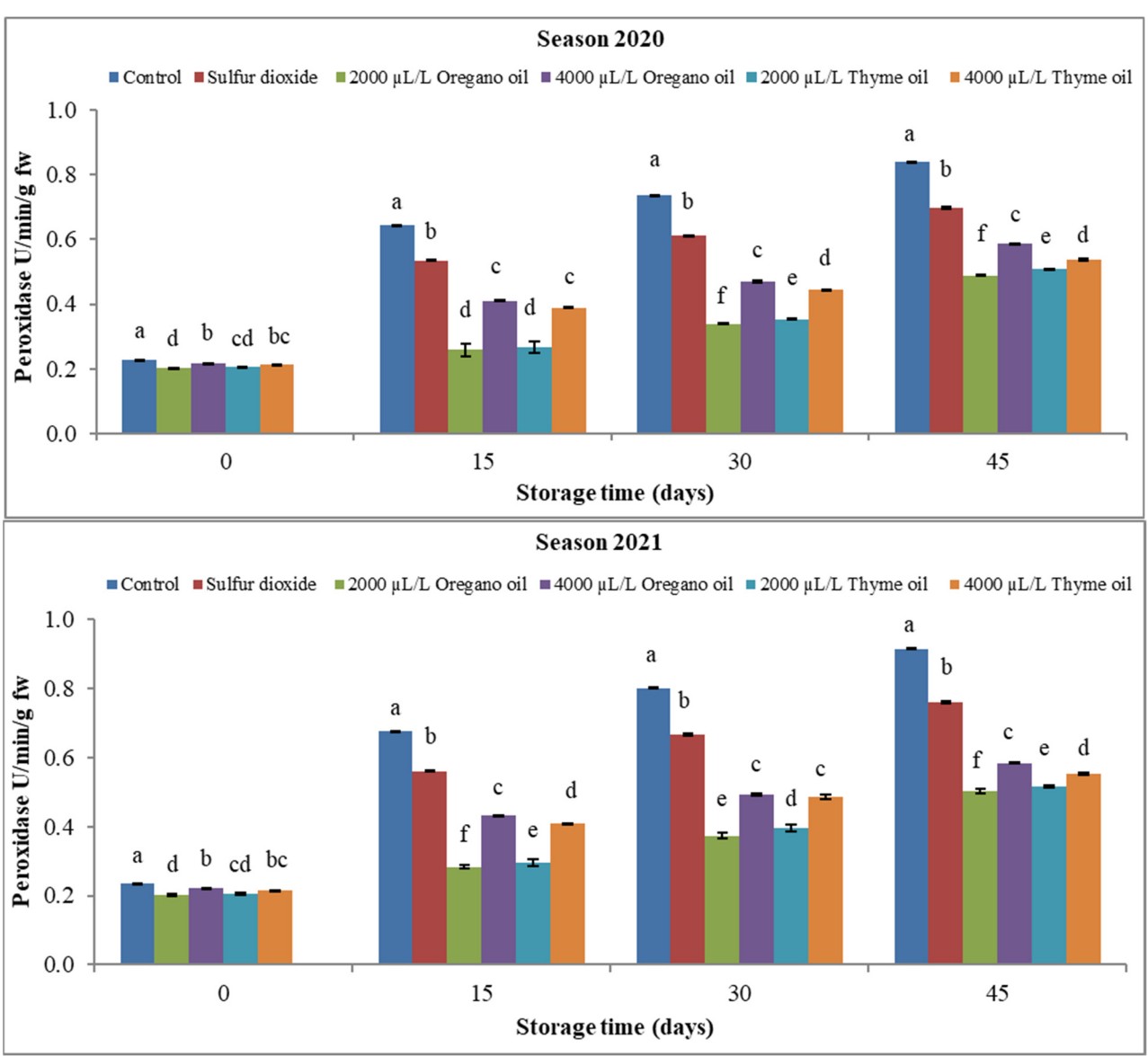

**Figure 17.** Effect of preharvest applications of oregano and thyme EOs on peroxidase activity of 'Flame Seedless' grapes during cold storage at 1 °C. Each value represents the mean ± SE of three replicates. Means followed by the same letters in each season and storage period are not significantly different according to Tukey's HSD test at $p \leq 0.05$.

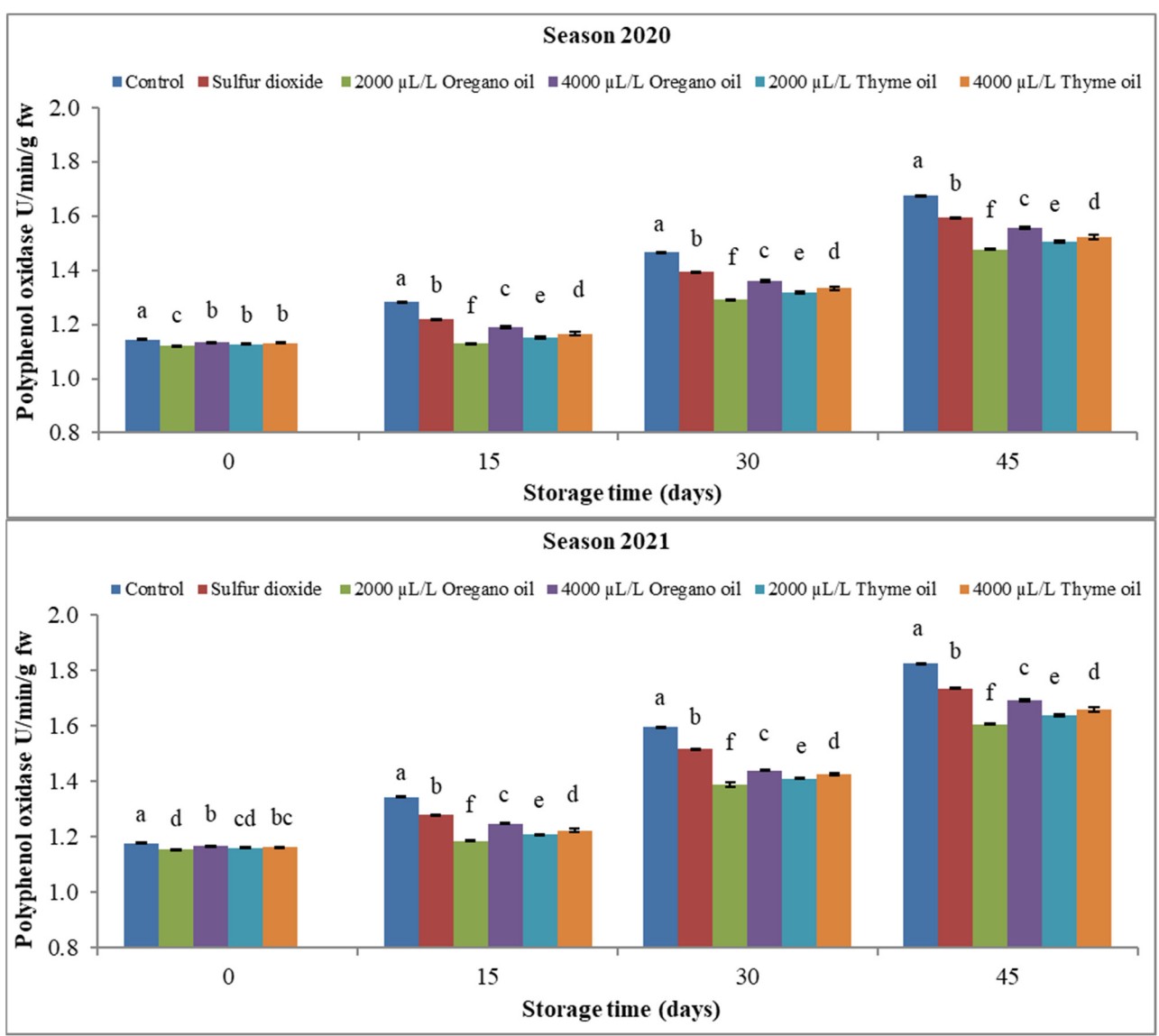

**Figure 18.** Effect of preharvest applications of oregano and thyme EOs on polyphenol oxidase activity of 'Flame Seedless' grapes during cold storage at 1 °C. Each value represents the mean ± SE of three replicates. Means followed by the same letters in each season and storage period are not significantly different according to Tukey's HSD test at $p \leq 0.05$.

### 3.3.2. Pectin Methylesterase (PME)

Pectin methylesterase activity showed the same trend as POX and PPO enzymes, showing increase with extending storage period (Figure 19). After 45 days of cold storage, the highest reduction in PME activity was reported for oregano essential oil treatments followed by thyme essential oil treatments and sulfur dioxide treatment, respectively. The highest PME activity was recorded under control.

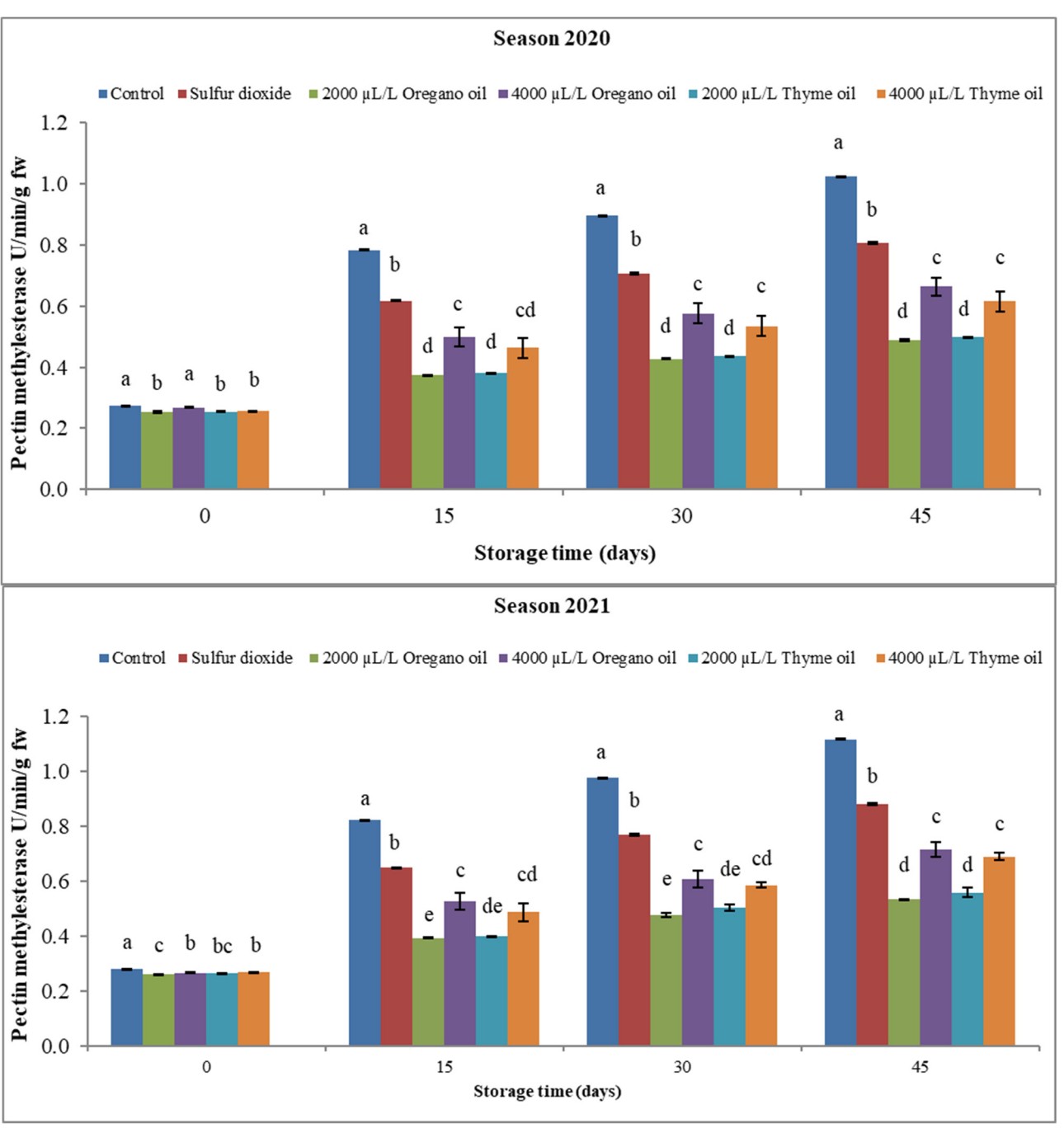

**Figure 19.** Effect of preharvest applications of oregano and thyme EOs on pectin methylesterase activity of 'Flame Seedless' grapes for 45 days cold storage at 1 °C. Each value represents the mean ± SE of three replicates. Means followed by the same letters for each season and storage period are not significantly different according to Tukey's HSD test at $p \leq 0.05$.

*3.4. Effect of Preharvest Applications of Oregano and Thyme Essential Oils on Shelf-Life per Day of 'Flame Seedless' Grapes during 45 Days of Cold Storage*

Shelf-life was enhanced due to oregano and thyme EO treatments compared to sulfur dioxide and control treatments (Figure 20). The longest shelf-life was found under oregano EO treatment at 2000 μL/L in both seasons, extending clusters' shelf-life up to about 6 days compared to about 2 and 2.5 days under control and sulfur dioxide treatments, respectively.

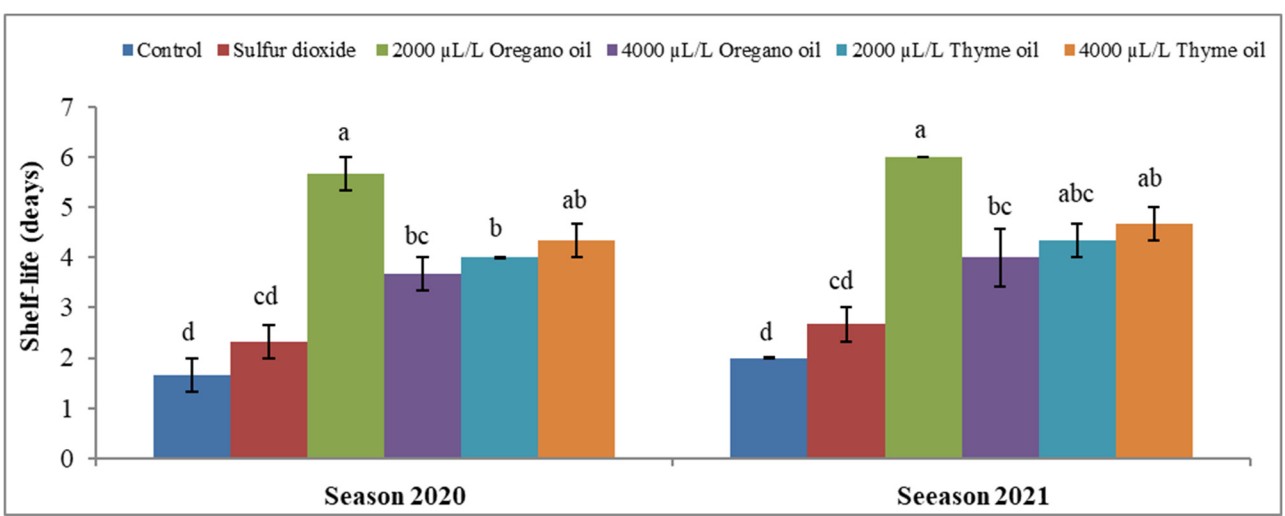

**Figure 20.** Effect of preharvest applications of oregano and thyme EOs on shelf-life of 'Flame Seedless' grapes after 45 days cold storage at 1 °C. Each value represents the mean ± SE of three replicates. Means followed by the same letters in each season are not significantly different according to Tukey's HSD test at $p \leq 0.05$.

*3.5. Principal Component Analysis (PCA)*

The goal of using PCA was to obtain a broader picture of the effect of the treatments on berries' physical and chemical properties, and the enzyme activities of the 'Flame seedless' grapevine.

With regard to the PCA (Figure 21), the score plot showed that all treatments affected the fruit's characteristics during both seasons. After 45 days of storage, the most pronounced effects were recorded for oregano EO at 2000 μL/L, thyme EO at 2000 μL/L, thyme at 4000 μL/L, and oregano at 4000 μL/L, followed by the effect of thyme 4000 μL/L and oregano at 4000 μL/L after 30 days of storage during the first season (Figure 21A). These treatments, particularly oregano at 200 μL/L after 45 days of storage, were mainly related to changes in PPO, weight loss after storage, and TSS, whereas oregano at 2000 μL/L after 45 days, followed by thyme at 4000 μL/L and oregano at 4000 μL/L after 30 days, were more effective on anthocyanins. On the other hand, treatments of oregano at 4000 μL/L, thyme at 4000 μL/L, thyme at 2000 μL/L, oregano at 2000 μL/L, after 30 days of storage, and thyme at 2000 μL/L at harvest, were mainly related to lower quality characteristics in terms of visual appearance and total acidity of grape berries. Other treatments such $SO_2$, control (untreated), after 15 days, $SO_2$ after 30 days, and $SO_2$ after 45 days were mainly effective on POX, TSS/Acid ratio and the percentage of decay after storage (gray mold). Principle component 1 and 2 accounted for 95.04% during the first season.

In the second season, the same treatments, oregano at 2000 μL/L, thyme at 2000 μL/L, thyme at 4000 μL/L, and oregano at 4000 μL/L, had the most pronounced effect after 45 days of storage, followed by the effect of thyme at 4000 μL/L and oregano at 4000 μL/L after 30 days of storage (Figure 21B). These treatments were mostly related to changes in PPO, weight loss after storage, and TSS, whereas oregano at 2000 μL/L after 45 days, followed by thyme 4000 μL/L and oregano 2000 μL/L after 30 days, were more effective on anthocyanins. On the other hand, the treatments thyme at 4000 μL/L, thyme at 2000 μL/L, oregano at 2000 μL/L, after 15 days of storage, thyme at 4000 μL/L, and thyme at 2000 μL/L at harvest were mainly related to lower quality characteristics in terms of visual appearance and total acidity of grape berries. Other treatments such as $SO_2$ after 15, 45, and 30 days, control (untreated) after 30 and 45 days of storage were mainly effective on POX, TSS/Acid ratio, decay after storage index, and the percentage of decay after storage (gray mold). Principle component 1 and 2 accounted for 95.28% during the second season.

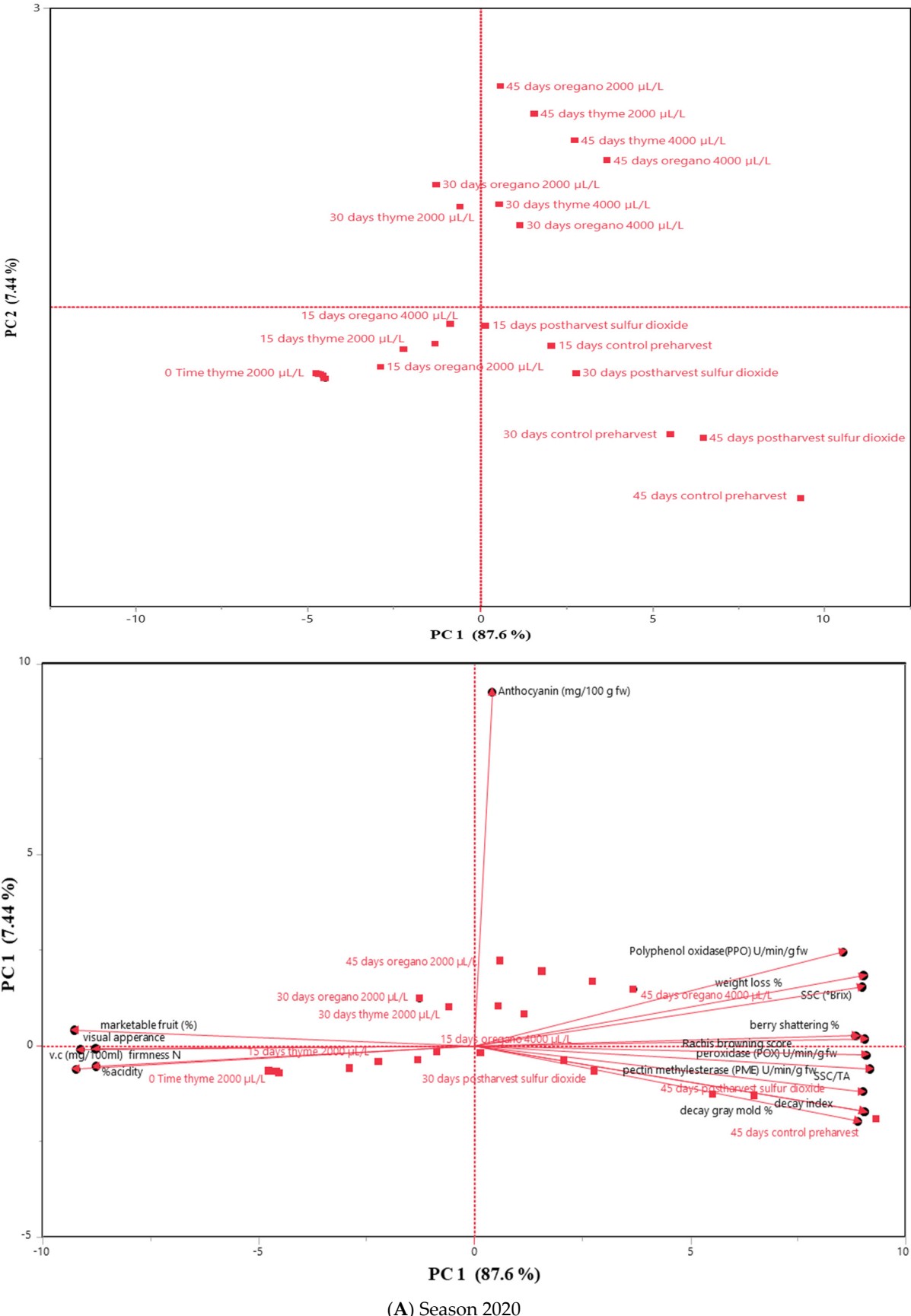

(**A**) Season 2020

**Figure 21.** *Cont.*

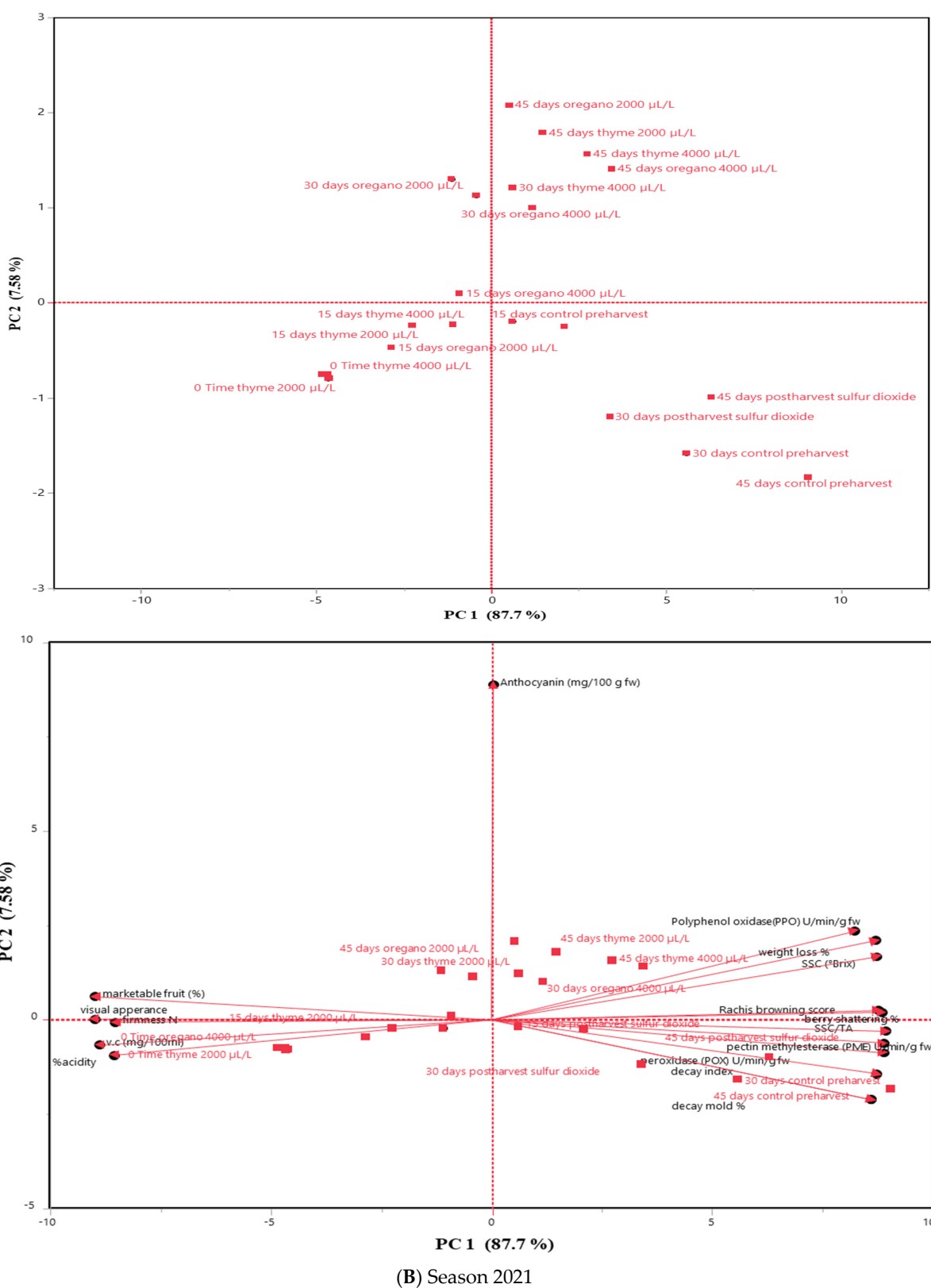

**Figure 21.** Principal component analysis (PCA) showing the score and loading plots of preharvest applications of oregano and thyme EOs during cold storage periods at 1 °C on 'Flame Seedless' grapes and fruit characteristics during the 2020 (**A**) and 2021 (**B**) seasons. Values are the means of three replicates (n = 3).

### 3.6. Economic Viability

The cost of various treatments shows that oregano and thyme essential oils treatment was less expensive than using sulfur dioxide pads (Figure 22). Furthermore, thyme had the lowest cost at each concentration.

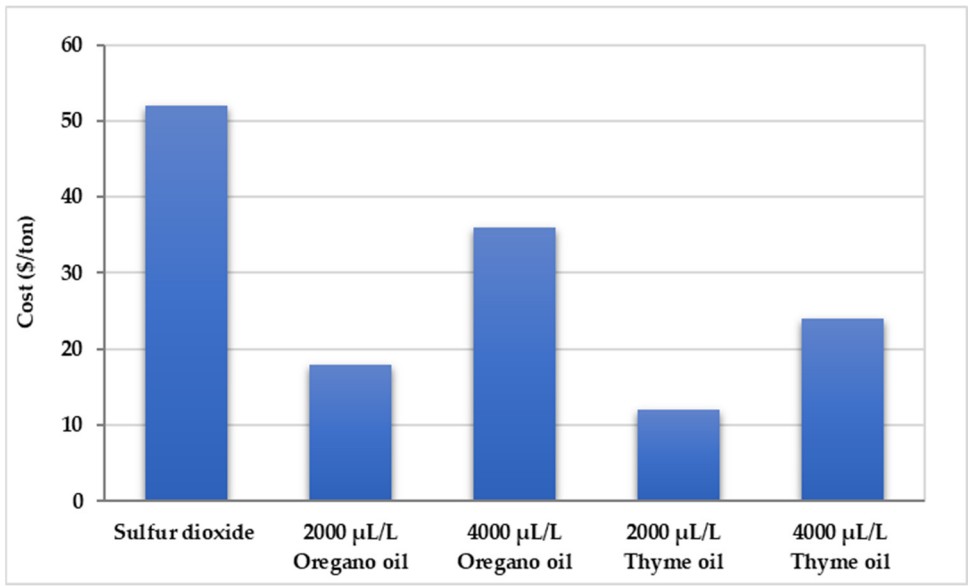

**Figure 22.** The cost per ton of preharvest applications of oregano and thyme essential oils compared to sulfur dioxide of 'Flame Seedless' grapes.

## 4. Discussion

*4.1. Effect of Preharvest Applications of Oregano and Thyme Essential Oils (EOs) on Physical Properties of 'Flame Seedless' Grapes during 45 Days of Cold Storage*

In global food production, natural alternatives to synthetic fungicides are desperately needed. In recent years, there has been a lot of interest in finding reasonably safe bio-fungicides, such as essential oils (EOs), to manage postharvest diseases while maintaining grape quality. EOs demonstrated good effects in delaying or decreasing the effects of environmental impact on fruit visual appearance [27]. Weight loss, decay, rachis browning, berry shattering, and berry firmness are important indicators of table grape bunch quality during cold storage. Our results indicated that preharvest application of oregano and thyme EOs in general reduced water loss and rachis browning, and controlled decay and gray mold incidence compared to the control and even sulfur dioxide treatment. Meanwhile, it maintained higher firmness, which consequently led to higher marketability and a longer shelf-life. Although oregano and thyme EO treatments at 2000 or 4000 μL/L were effective in reducing the deterioration of physical parameters of treated bunches at the end of cold storage, oregano and thyme EO treatments at 2000 μL/L were the most effective.

Approximately 5% weight loss was found to be the normal acceptable limit for table grapes during storage [58]. Our results indicated that 45 days after storage, both oregano and thyme EOs resulted in a significant decrease in weight loss ranging from 4.06 to 5.26% compared to untreated (7.28%) and sulfur dioxide (5.33%) treatments, as an average of both seasons. The progress of weight loss could be partly attributed to the increase in the berry's metabolic activity associated with tissue senescence over long storage times, which is slowed after the application of essential oils [59]. Such reduction in water loss was reported by Krasniewska et al. [60] due to the application of essential oils together with edible coatings. Furthermore, Behshti et al. [61] found that the application of marjoram essential oil decreased the percentage loss in fresh weight significantly and increased the storage life of grapes. Similar results were reported by Burt [27], Nabifarkhani et al. [59],

Rattanapitigorn et al. [62], Valero et al. [63], Abdolahi et al. [64], Abdollahi et al. [65], and Khan et al. [66].

Decay and gray mold caused by *Botrytis cinerea* is the most economically important postharvest disease of table grapes. Our results indicated that oregano and thyme EOs significantly reduced decay and gray mold percentages compared to the control and $SO_2$ treatments, with the lowest percentage at oregano EO treatment at 2000 μL/L followed by thyme EO at the same concentration, in both seasons after 45 days of storage. The antifungal activity of EOs was proved in various studies on grapes [66–68], apples [69], and strawberries [70]. Carvacrol and thymol have been shown to have fungicidal activity, and EOs rich in these components have been shown to have high inhibitory activity against a variety of postharvest pathogens [36]. GC-MS analysis of oregano and thyme essential oils revealed 25 and 27 components (Figure 1), respectively, and both are rich in monoterpenes, which may have fungicidal properties that protect fruit against qualitative and quantitative losses, thereby improving fruit storability [71]. The main compounds in oregano essential oil are carvacrol (47.14%), α-terpineol (16.71%), and *p*-cymene (11.47%), while thymol (41.94%), γ-terpinene (17.03%) and *p*-cymene (16.22%), are the main compounds in thyme essential oil. Gray mold did not display any mycelial growth in the presence of lemon and oregano essential oils, according to Vitoratos et al. [39], and mycelial growth of *Monilinia fructicola* was effectively reduced due to thyme essential oil application, Svircev et al. [72]. Furthermore, thyme oil, specifically 0.2 mL/mL, had a significant antifungal effect on gray mold in vitro [73] and oregano essential oil inhibited gray mold in tomato [39]. EOs are natural compounds containing a complex mixture of odorous and volatile constituents that have antibacterial and antifungal activities through to inhibit conidial germination, resulting in the fungus being killed [74,75]. Further mechanisms of EOs' antifungal action were suggested; some components of EOs may disrupt the cell membrane by cross-linkage reactions, causing leakage of electrolytes and depletion of amino acids and sugars, whereas other components may selectively be inserted into the lipid-rich portion of the cell membrane, thereby disturbing membrane function which results in microbial cell energy loss [76,77]. Moreover, EOs are primarily conjugated with phenolic compounds, which interact with membrane proteins and alter microbial cell permeability, thereby lowering the incidence of infection and decay [40,78].

The development of rachis browning during postharvest storage has been reported in table grapes [79] and other crops such as apricot [80] and strawberry [81,82]. This study indicated that oregano and thyme EOs resulted in significant reduction in RBI and BS percentage in grapes compared to untreated and sulfur dioxide treatments across all storage periods. Browning incidence is primarily due to the cumulative water loss that occurs after harvest, but it may also be caused by phenolic compound oxidation by polyphenol oxidase (PPO) processes [83]. Berry shattering increased with extending storage period and such increase is in conjunction with water loss that triggers ethylene production and formation of an abscission layer at the distal end of the pedicel of a berry [84–86]. It has been additionally proposed that the increased berry shatter during storage indicates that the pedicel and stalk of the cluster behave in a climacteric process, exhibiting respiration and ethylene peaks [86]. In the current study, oregano and thyme EOs, particularly at low concentrations (2000 μL/L), reduced RBI and BS percentages by delaying PPO enzyme activity and decreasing weight loss percentages when compared to untreated and sulfur dioxide treatments. Abdolahi et al. [64] demonstrated the role of EOs in reducing RBI in grapes by reporting that thyme and summer savoury EOs had significant efficacy on fruit quality parameters such as weight loss, grape berry shrinkage, and rachis browning. Furthermore, Abdollahi et al. [65] demonstrated that thyme oil reduced rachis browning while having no significant adverse effect on the flavor of table grapes.

Consumers value table grape firmness, and excessive softening can lead to postharvest decay or consumer refusal. The application of oregano and thyme EOs in Flame seedless grapes retarded grape berry softening, recording the highest firmness under the preharvest applications 2000 μL/L of oregano followed by thyme essential oil at the same concentra-

tion. Previous research has found that EOs have a positive effect on firmness. Hassani et al. [81] found that applying thyme essential oil to apricot fruit resulted in significant firmness retention. In addition, Gago et al. [87] reported that treating pear with lemongrass essential oil postponed the change in fruit firmness during cold storage. Similarly, Das et al. [88] indicated that coating stored fruits with essential oil is effective in maintaining firmness. The decrease in firmness could be attributed to cell wall degradation in the middle lamella caused by the breakdown of insoluble protopectin into soluble pectin [89], as well as fungal infection attack, which causes hydrolyzed pectin and cell wall breakdown [90,91]. Furthermore, coating berries with EOs can result in a low oxygen and high carbon dioxide atmosphere within the coating surrounding the fruit, reducing the rate of fruit respiration [92,93]. Low oxygen concentrations in the environment surrounding the fruits may reduce the activities of enzymes such as pectin esterase, pectin methylesterase esterase, and polygalacturonase enzymes, which accelerate the depolymerization of pectin and other pectic substances responsible for structural rigidity [94]

*4.2. Effect of Preharvest Applications of Oregano and Thyme Essential Oils on Physio-Biochemical Properties of 'Flame Seedless' Grapes during 45 Days of Cold Storage*

AsA is a non-enzymatic antioxidant that plays an important role in protecting fruit cells during storage by reducing oxidative damage caused by reactive oxygen species (ROS) [95]. Furthermore, SSC, TA, and their ratio are the major components associated with flavor properties and are regarded as key indicators of fruit senescence [96]. The AsA content of 'Flame Seedless' grapes decreased gradually during storage under all treatments, indicating a reduced ability to prevent oxidative damage by a rapid conversion of L-ascorbic acid into dehydroascorbic acid, which is also oxidized to diketogulonate in the presence of oxidation enzymes [97], and this is associated with the occurrence of physiological disorders [98]. A continuous reduction in AsA and TA, and a continuous increase in SSC and SSC/TA with the progress of cold storage periods, was reported by El-Abbasy et al. [99]. The degradation of AsA and TA in the untreated and sulfur dioxide treatments was the highest and this could be attributed to exacerbated physiological disorders and a high respiration rate [100,101]. The increase in SSC during storage is due to a consequence of cell wall degradation in the fruit and higher carbohydrate hydrolysis into sugars by hydrolytic enzymes causes an increase in water soluble galacturonic acids [102]; this also could be attributed to the higher rate of evaporation and respiration from the grape surface [85,103,104]. The content of AsA and TA in grapes treated with oregano and thyme EOs were higher at the end of the storage period especially at low concentration (2000 µL/L) compared to untreated sulfur dioxide treatments. The positive effect of oregano and thyme EOs could be attributed to the antioxidant capabilities of EOs coatings, which decrease oxygen diffusion and oxidation of superoxide and hydroxyl radicals, and decrease the loss of water, reduce microbial growth, and cause a slower respiration rate and conversion of organic acid into sugars during the metabolic processes of berry [105–107]. Furthermore, the ability of EOs to minimize water loss from the fruit surface and lower carbohydrate hydrolysis into sugars was associated with a reduction in fruit SSC [40]. Our findings are consistent with those of Salimi et al. [108], who found that using basil and wild mint essential oils preserved the AsA and TA contents of grapes during cold storage. Das et al. [88] reported that essential oil coating was effective in maintaining the organoleptic and nutritional qualities of stored grapes. Similar results were obtained by Hassani et al. [80] in apricot fruit and Geransayeh et al. [109] in strawberry.

Cold storage of grapes could reduce the continuous loss of biologically active anthocyanins as well as the development of brown color. The degradation of berry peel anthocyanin contents is linked to an increase in PPO and POX enzyme activities [110]. Anthocyanins may also be converted into anthocyanidins and, eventually, quinones, which condensate to form brown polymers [111]. As a result, significantly higher anthocyanin contents in grapes treated with oregano and thyme EOs could be attributed to anthocyanidin suppression, delayed phenolic oxidation, and reduced activities of POX and PPO

enzymes [110]. Furthermore, the increase in total anthocyanin content during storage in EO-treated grapes can be attributed to the activation of major anthocyanin precursors such as cyanidin, peonidin, and acylated derivatives in berry skin and pulp [112]. Components of EOs, such as carvacrol and thymol, have been shown to increase antioxidant levels (polyphenols, flavonoids, and anthocyanins) and oxygen absorbance capacity in plant tissues, including in enzymatic and non-enzymatic systems, which increase the oxidation of oxygen radicals and hydroxyl radicals in fruit tissue [113,114]. Several studies reported that EOs had significant efficacy on total anthocyanin, such as Nabifarkhani et al. [59], who demonstrated that thyme oil significantly affects fruit water retention and thus preserves anthocyanin compared to sweet cherry control samples. Likewise, Behshti et al. [61] discovered that grape berries treated with marjoram oil had higher total anthocyanin content. Sulfur dioxide treatment of table grapes resulted in discoloration of the skin of the 'Redglobe' berry due to sulfur dioxide accumulation in the pedicels [5]. In this study, the degradation of anthocyanin in untreated and sulfur dioxide treatments after 30 and 45 days of cold storage may be due to berry senescence attributed to physical and chemical deterioration such as decay incidence, browning, loss of AsA, and increased PPO enzyme activity.

### 4.3. Effect of Preharvest Applications of Oregano and Thyme Essential Oils on Enzyme Activity of 'Flame Seedless' Grapes during 45 Days of Cold Storage

Peroxidase (POX) and polyphenol oxidase (PPO) activities gradually increased during cold storage in all treatments. These increases might be attributable to the breakdown of cell structure at senescence [115], where diphenols can act as reducing substrates in the POX and PPO reaction, and another oxidoreductase enzyme is involved in enzymatic browning [116]. Enzymatic browning, which occurs primarily as a result of phenolic compound oxidation, causes color deterioration in fruits and contributes to quality loss as well as significant economic loss due to consumer unacceptability [117]. PPO is the primary enzyme responsible for the browning reaction because it can catalyze the hydroxylation of monophenols to o-diphenols, which can then be oxidized to o-quinones, which cause tissue browning [118]. POX is also an important enzyme in the phenylpropanoid metabolic pathway, which promotes synthesis [119]. Biles and Martyn [120] explain the role of POX in damaged tissues, describing its involvement in cell wall cross-linking and pathogen-infected tissues. As a result, increasing POX and PPO activity would reflect the progression of tissue damage during storage, whereas lower POX and PPO activity would indicate less tissue damage. During cold storage, oregano and thyme EOs at 2000 and 4000 μL/L significantly reduced the activities of POX and PPO enzymes compared to untreated and sulfur dioxide treatments. EOs of oregano and thyme at 2000 μL/L had the best results in reducing POX and PPO activities. The inhibition of POX and PPO activities by oregano and thyme EOs may be due to abundant phenolic compounds and increased antioxidant capacity, which leads to the removal of free radicals [113,114,121], It may also be related to a delay in senescence and to the delayed oxidation of phenol substances through cold storage [64]. Thymol applied to the fruits in the previous study was shown to inhibit POX activity compared to control [66]. Similarly, Jin et al. [122] discovered that essential oil treatments applied after harvest (carvacrol, anethole, cinnamaldehyde, linalool, perillaldehyde, and cinnamic acid) improved raspberries' antioxidant capacity.

Pectin methylesterase (PME) activity continued to increase during cold storage in all treatments. The increase in PME during storage is associated with the ripening process, which occurs mainly because of the degradation of the middle lamella of the cell wall [89] and is also probably due to fungal infection [90]. Treating grapes with oregano and thyme EOs, especially at the lower concentration (2000 μL/L), had the lowest activity of PME enzyme compared to the untreated and sulfur dioxide treatments during cold storage. EOs may modify the atmosphere surrounding the fruit, which reduces the respiration rate and inhibits the formation of ethylene hormone [92]. Coating with EOs showed the lowest oxygen consumption rate, which plays a key role in the ripening process and reduces the

activity of PME [94,123]. The role of EOs in reducing PME activity and keeping cell wall integrity was clear in terms of its effect on maintaining higher fruit firmness and reduced incidence of decay and gray mold.

### 4.4. Economic Viability

Results indicated that using both essential oils was more economical than sulfur dioxide treatment. Taking into account the recommended concentration (2000 L/L), the cost per ton for oregano is 34% that of sulfur dioxide and 23% for thyme.

### 5. Conclusions

Preharvest application of oregano and thyme EOs at 2000 and 4000 μL/L proved effective in maintaining grape quality during cold storage. When compared to sulfur dioxide, the low concentrations of both EOs significantly reduced decay and gray mold incidence, weight loss, rachis browning index, and berry shattering percentage, and showed the highest marketable fruit percentage, firmness, and visual appearance score. Furthermore, retained berry contents of ascorbic acid and titratable acidity (TA) were achieved with slowing the increases in soluble solids content (SSC) and SSC/TA ratio. In addition, PPO, POX, and PME enzyme activities were also inhibited. The results show that preharvest application of oregano and thyme EOs at 2000 L/L two days before harvest can be an eco-friendly, cost-effective, and safe alternative to the commonly used synthetic fungicide sulfur dioxide in extending storage life while maintaining fruit's physio-biochemical attributes.

**Author Contributions:** Conceptualization, A.F.A.E.-K.; methodology, U.K.E.-A., A.F.A.E.-K. and M.A.A.-H.; software, A.F.A.E.-K. and M.A.A.-H.; validation, U.K.E.-A., A.F.A.E.-K. and M.A.A.-H.; formal analysis, A.F.A.E.-K. and M.A.A.-H.; investigation, A.F.A.E.-K. and M.A.A.-H.; resources, A.F.A.E.-K., H.M.H.-V. and M.A.A.-H.; data curation, A.F.A.E.-K., A.R.E.-S. and M.A.A.-H.; writing—original draft preparation, A.F.A.E.-K. and M.A.A.-H.; writing—review and editing, A.F.A.E.-K., A.R.E.-S. and H.M.H.-V.; visualization, A.F.A.E.-K. and A.R.E.-S.; supervision, U.K.E.-A. and A.F.A.E.-K.; project administration, U.K.E.-A. and A.F.A.E.-K. All authors have read and agreed to the published version of the manuscript.

**Funding:** This research received no external funding.

**Data Availability Statement:** Data will be made available upon reasonable request from corresponding author.

**Acknowledgments:** The authors gratefully thank the owner of the grape orchard and grape packing house, and his staff for providing all the required materials to do this research. The authors' appreciation also extends to Postgraduate Sector, Faculty of Agriculture, Tanta University, Tanta, 31527, Egypt, Shamel M. Alam-Eldein, and Islam F. Hassan for their excellent assistance.

**Conflicts of Interest:** The authors declare no conflict of interest.

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
