# Peer review of "Effectiveness of Oregano and Thyme Essential Oils as Alternatives for Sulfur Dioxide in Controlling Decay and Gray Mold and Maintaining Quality of ‘Flame Seedless’ Table Grape (Vitis vinifera L.) during Cold Storage"

_agronomy, doi:10.3390/agronomy13123075_

Round 1
Reviewer 1 Report
Comments and Suggestions for Authors
1- Line 38-40, “in 2020, the global total harvested area of grapes 38 was 6.95 million hectares, with a total production of 78 million tons, while Egypt's total 39 harvested area was 71.9 hectares, with a total production of 1.6 million tons” Could the author check very well this information, because how can Egypt have 71.9 hectares and produce 1.6 million tons?
2- Authors may point well in introduction the gap of knowledge that their research will fill.
3- Line 42-45, authors are encouraged to give example of valued active compounds.
4- Line 80, fruits physical and chemical properties, it's not very clear, what do the authors mean by chemical properties of fruits? Also in Line 760, chemical properties, we suggest that the authors to consider systematically replacing this expression throughout the text with “physio-biochemical attributes” as the authors mentioned in line 187 or more better expression.
5- As the reader will find it difficult (number are too small to read in Fig. 1) to make the correspondence on the basis of color, can the authors present this composition in the form of a composition table rather than a chart?
6- Line 121, for which country?
7- The author can draw up a diagram to simplify the presentation of the experimental procedure to the reader.
8- Authors must also justify why they chose essential oil concentrations of 0, 2000 and 4000 µL/L and storage periods of 0, 15, 30 and 45 days.
9- Line 210, author can give details of which tissue part has been considered for every procedure of enzyme measurement for example, whole fruit, flesh or peel?
10- Authors must choose: µL and mL or µl and ml, and apply it in whole text.
11- Figure 17, authors should write well the small letter “d” in bend.
12- Italicize micro-organism names throughout the text, e.g., Line 588 Botrytis cinerea
13- The effect season and PCA result is not sufficiently taken into account in the discussion. In comparison to other reports, authors should emphasize these points.
14- In 2.2. Plant materials and experimental procedure, Can the authors indicate the month of the year or the environmental parameter at the time of spraying with oregano or thyme essential oils?
Comments on the Quality of English LanguageModerate editing of English language required
Author Response
The point-by-point response to the reviewer's comments are enclosed in the attached file.

Reviewer 2 Report
Comments and Suggestions for Authors
The manuscript «Effectiveness of Oregano and Thyme Essential Oils as Alterna- 2 tives for Sulfur Dioxide in Controlling Decay, Gray Mold and 3 Maintaining Quality of ‘Flame Seedless’ Table Grape (Vitis vi- 4 nifera L.) During Cold Storage» of El-Abbasy U. K. et al. described the effect of oregano and thyme EO on some physiological parameters and oxidant enzymes activity during cold storage.
Some questions and remarks listed below:
1. Why the oregano and thyme EO were used? In introduction partly described the antimicrobial effect. But some fatty acids and their composition also have antibacterial capabilities. In introduction needs to intensify the chosen of oregano and thyme EO.
2. Will be another plant’s EO, like sunflower or rape oil has the same positive effect? If such kind of data are available it can be added in Introduction section
3. In section 2.3.3 PPO activity, in description of the method was missed the absorbance of optical density (line 226) and volume of 0.1M sodium phosphate buffer that was added into solution (line 225). Please, to add this information
4. In Discussion section will be better to add some short information about economic viability of using the oregano and thyme EO in comparative to SO2.
5. What kind of products made from grape after storage? How the addition of oregano and thyme EO can affect on the products after storage the grape? This information also can be interesting for underline the eco-friendly and safety of using the oregano and thyme EO
Author Response

(The authors gave the same response as above.)

Round 2
Reviewer 1 Report
Comments and Suggestions for Authors
The authors have taken account of the main comments. In particular, they have highlighted the gaps in the introduction, as suggested, and have drawn up an experimental procedure that will improve readability. I have no additional comments.
Reviewer 2 Report
Comments and Suggestions for Authors
Authors added information according to mine recomendations